# Proactive Agent: Shifting LLM Agents from Reactive Responses to Active Assistance

**Yaxi Lu**[1], **Shenzhi Yang**[2], **Cheng Qian**[1], **Guirong Chen**[2], **Qinyu Luo**[1], **Yesai Wu**[1],
**Huadong Wang**[1], **Xin Cong**[1], **Zhong Zhang**[1], **Yankai Lin**[2], **Weiwen Liu**[3], **Yasheng Wang**[3],
**Zhiyuan Liu**[1*], **Fangming Liu**[4*], **Maosong Sun**[1]

[1] Department of Computer Science and Technology, Tsinghua University
[2] Gaoling School of Artificial Intelligence, Renmin University of China
[3] Huawei Noah's Ark Lab, [4] Peng Cheng Laboratory
lyx23@mails.tsinghua.edu.cn, mrlyk423@gmail.com,
liuzy@tsinghua.edu.cn, fangminghk@gmail.com

## Abstract

Agents powered by large language models have shown remarkable abilities in solving complex tasks. However, most agent systems remain reactive, limiting their effectiveness in scenarios requiring foresight and autonomous decision-making. In this paper, we tackle the challenge of developing proactive agents capable of anticipating and initiating tasks without explicit human instructions. We propose a novel data-driven approach for this problem. Firstly, we collect real-world human activities to generate proactive task predictions. These predictions are then labeled by human annotators as either accepted or rejected. The labeled data is used to train a reward model that simulates human judgment and serves as an automatic evaluator of the proactiveness of LLM agents. Building on this, we develop a comprehensive data generation pipeline to create a diverse dataset, ProactiveBench, containing $6,790$ events. Finally, we demonstrate that fine-tuning models with the proposed ProactiveBench can significantly elicit the proactiveness of LLM agents. Experimental results show that our fine-tuned model achieves an F1-Score of $66.47\%$ in proactively offering assistance, outperforming all open-source and close-source models. These results highlight the potential of our method in creating more proactive and effective agent systems, paving the way for future advancements in human-agent collaboration.

## 1 Introduction

The emergence of large language models (LLMs) like ChatGPT (OpenAI, 2022) has significantly advanced the development of autonomous agent (Chen et al., 2023; Hong et al., 2023; Zhang et al., 2023; Wu et al., 2023). These LLM-based agents can understand human instructions, make plans, explore environments, and utilize tools to solve complex tasks (Wu et al., 2023; Li et al., 2023) and have shown substantial potential in various applications such as robotics (Firoozi et al., 2023), personal assistants (Li et al., 2024), and process automation (Ye et al., 2023b).

Currently, most existing LLM-based agents predominantly work in the reactive paradigm: they require explicit human instructions to initiate task completion and remain dormant in terms of providing services until prompted by user instructions (Ouyang et al., 2022). This paradigm limits their capacity for proactive assistance and autonomous service provision in the absence of direct human instructions. We argue that **LLM-based agents should be proactive, capable of autonomously initiating tasks by understanding and responding to their environment**. For instance, as illustrated in Figure 1, the reactive agent should wait for explicit instructions from the user to execute tasks such as "show unread emails" or "schedule a meeting with John". In contrast, a proactive agent would predict its task automatically by noticing an email from John suggesting a meeting and automatically offering to schedule it. This ability of context awareness (Schilit & Theimer, 1994) enables the proactive agent to interpret signals and proactively propose and execute tasks without

---

*Corresponding authors.

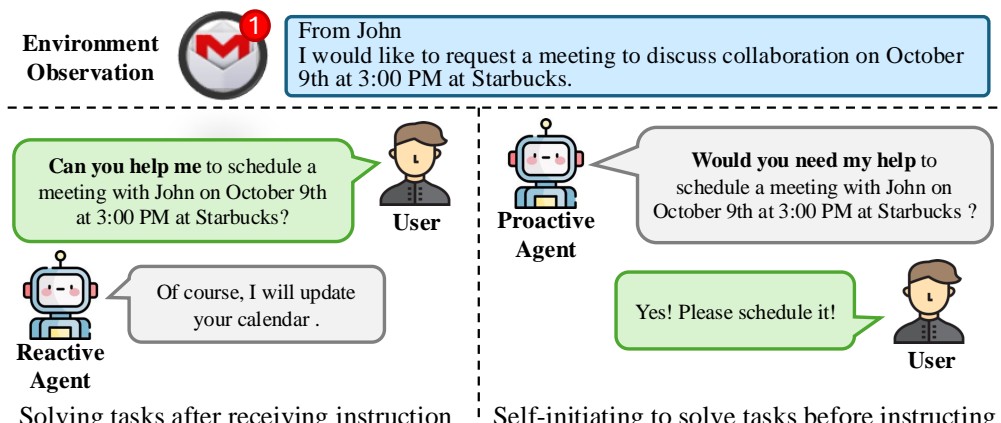

Figure 1: Comparison of agent systems with two types of human-agent interaction. Reactive agents passively receive user queries and then generate responses. Proactive agents infer tasks based on environmental observations and propose possible assistance requests accordingly.

explicit human instructions. Thus, it not only significantly reduces the cognitive burden on the user but also identifies latent needs not explicitly articulated by humans. Consequently, the proactive agent could provide more comprehensive and seamless services to the user.

In this work, we propose a novel data-driven formalization for developing a proactive agent that anticipates user needs and takes the initiative by suggesting tasks or providing information without explicit requests. Our approach centers on constructing ProactiveBench, allowing us to evaluate and enhance the agent's proactive behavior. Firstly, we collect real-world human activity data in three settings: coding, writing, and daily life. This includes but is not limited to, user keyboard and mouse inputs, clipboard content, browser activity, etc. Then, we build an LLM-driven gym to generate events that reflect the raw real-world contexts

| Subsets | Scenarios | Entries (Tr/Ts) |
|---|---|---|
| Agent Model | 136 | 6,790 / 233 |
| *Coding* | 46 | 2,275 |
| *Writing* | 46 | 2,354 |
| *Daily Life* | 44 | 2,161 |
| Reward Model | - | 1,640 / 120 |

Table 1: Statistics of the ProactiveBench, which includes three distinct settings: Coding, Writing, and Daily Life.

we collected. We obtain a total of 233 events across 12 scenarios as the test set of the ProactiveBench. To further refine the proactive behavior of the LLM-based agent, we construct various events and proactive tasks under synthetic contexts with the gym. By iterative generating more events and predictions, we obtain up to $6,790$ events as the train set of the ProactiveBench, as shown in Table 1. We fine-tune the LLaMA-3.1-8B-Instruct (Touvron et al., 2023; Meta, 2024) and the Qwen2-7B-Instruct (Bai et al., 2023) on this training set to refine their proactive behavior.

To automatically evaluate the proactiveness of LLMs, we train a reward model that achieves up to $91.80\%$ consistency with human judgments in terms of F1-Score. Using this model, we compared various language models on ProactiveBench. The results indicate that even the latest open-source models struggle to effectively predict proactive tasks. For instance, the LLaMA-3.1-8B model only achieved a $55.06\%$ F1-Score on ProactiveBench. In contrast, our fine-tuned model demonstrated significant improvements, achieving a $66.25\%$ F1-Score. Besides, our fine-tuned Qwen2-7B-Instruct model achieves $66.47\%$ F1-Score, outperforming all existing open-source and closed-source LLMs. This underscores the effectiveness of our data-driven approach in developing proactive agents, highlighting their potential to enhance user experiences across various applications.

## 2 RELATED WORKS

Recent advancement in large language models (OpenAI et al., 2023; Chowdhery et al., 2022; Touvron et al., 2023; Zeng et al., 2023) has shown great progress in complex reasoning, task plan-

ning (Wei et al., 2022; Gao et al., 2023; Yao et al., 2023b;a; Liu et al., 2023; Ye et al., 2023a;b), tool utilization (Schick et al., 2023; Qin et al., 2023b;c; Qian et al., 2024b), etc. Consequently, a growing number of agent systems have been developed to utilize these models to tackle diverse tasks like automatic web search (Qin et al., 2023a), software development (Qian et al., 2023; Chen et al., 2023), behavior simulation (Park et al., 2023). Despite these advancements, most current agents remain predominantly reactive, passively following human instructions without sufficient context awareness (Schilit & Theimer, 1994) to proactively meet user needs. These reactive agents typically wait for explicit user commands, which can lead to inefficiencies as task complexity increases. As a result, users must constantly provide specific inputs, hindering the flow of interaction. In response, several works have attempted to improve the proactivity of agents. For example, Xuan (Zhang et al., 2024) proposes proactive agent planning, where agents refine their tasks by actively seeking information to better understand user intentions. Cheng Kuang and Zhi Rui (Wu et al., 2024) study how to propose proactive support in text-to-sql and propose a novel metrics Area Under Delta-Burden Curve (AUDBC). Other studies (Wu et al., 2021; Chen et al., 2020; Wiethof et al., 2021; Qian et al., 2024a) focus on enabling multi-turn interactions to clarify ambiguous user instructions, which further increased cognitive load for the user. However, these works still require the user to give an initial query before interacting with the agent. Our approach takes a different direction by focusing on anticipating potential tasks based on monitoring user activities and environmental states, which allows the agent to proactively initialize the interaction and provide assistance.

To clarify, there are also previous works (Deng et al., 2023) that use the term "Proactive Agent" to describe their dialogue systems. However, most of these efforts (Bi et al., 2021; Ren et al., 2021; LI & LIAO; Liu et al., 2024) aim to enhance the helpfulness or quality of responses in a proactive manner, which differs from our focus on task anticipation and initiation.10814

## 3 METHODOLOGY

### 3.1 TASK DEFINITION

In our proposed proactive agent, which is distinct from traditional agent systems powered by large language models that rely on explicit user instructions, we investigate a new scenario where the agent autonomously predicts tasks users might assign, aiming to offer assistance proactively, as depicted in Figure 1. The proactive agent's mission is to give predictions based on the user's activities $A_t$, environmental events $E_t$, and state $S_t$, which can be formalized as:

$$P_t = f_\theta(E_t, A_t, S_t) \tag{1}$$

where $f_\theta$ represents the proactive agent, parameterized by $\theta$, and $P_t$ denotes the prediction about possible task at time $t$. It should be noticed that the prediction $P_t$ can be the predicted task or nothing if the agent believes that no task is needed. Specifically, user activities $A_t$ contain the user's interactions with the environment and the agent, like keyboard input or chatting with the agent. Environmental events $E_t$ contain the event that the proactive agent captured, ranging from receiving a new email to an application closed. Environmental state $S_t$ represents the state of the current environment, like the file system state or the content of opened web pages.

In our proactive agent framework, the objective is to maximize the user's acceptance rate of the proposed tasks. Given the user's historical activities $A_t$, current environmental state $S_t$, and the prediction proposed by the proactive agent $P_t$, the user makes a binary decision:

$$R_t = g(P_t, A_t, S_t) \tag{2}$$

where $R_t$ is a binary variable indicating acceptance ($R_t = 1$) or rejection ($R_t = 0$) of the prediction. To unify the handling of cases where the prediction $P_t$ contains no task and where it contains a task, we introduce an auxiliary variable $N_t$ that indicates whether the user needs assistance:

- $N_t = 1$ if the user needs assistance.
- $N_t = 0$ if the user does not need assistance.

The user's acceptance $R_t$ is then defined as:

$$R_t = \begin{cases} 1 & \text{if } (P_t \neq \emptyset \text{ and user accepts } P_t) \text{ or } (P_t = \emptyset \text{ and } N_t = 0) \\ 0 & \text{otherwise} \end{cases}.$$

In this way, if the prediction $P_t$ contains no task (i.e., the agent believes the user does not need assistance), we check the user's actual need for assistance $N_t$. If the user indeed does not require assistance ($N_t = 0$), this is marked as acceptance ($R_t = 1$). Conversely, if the user requires assistance ($N_t = 1$), this is marked as rejection ($R_t = 0$). Our proactive agent aims to maximize the expected acceptance rate of the proposed tasks:

$$\max_\theta \mathbb{E}[R_t]. \tag{3}$$

## 3.2 PIPELINE OVERVIEW

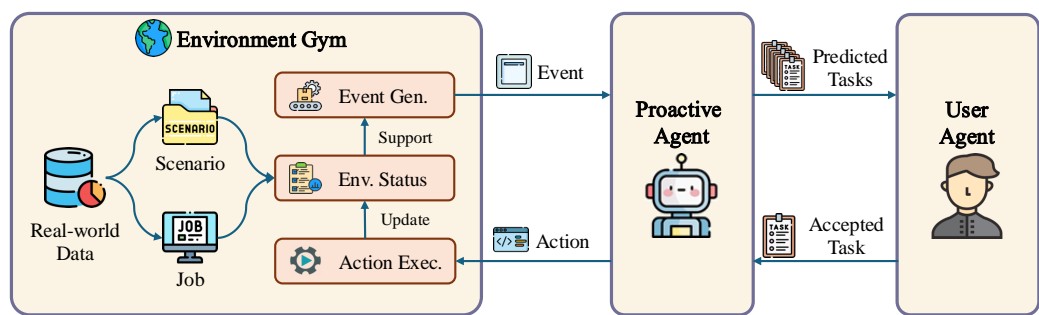

Figure 2: Overview of the data generation process. Taking daily life as an example, this process includes modules such as the initial scenario and job setup, events generation, proactive prediction, user judgment, and action execution.

To enhance the proactive capabilities of our large language model-powered agent, we adopt a data-driven approach by building an automatic data generation pipeline. This pipeline simulates user activities and responses to the tasks predicted by the proactive agent across various scenarios. Once a prediction is accepted, we simulate the agent performing the task by interactively generating new events within the simulated environment. Subsequently, new user activities are created based on historical events, allowing the proactive agent to generate further predictions. Through this pipeline, models can learn when to generate predictions and which predictions are likely to be accepted by users. Specifically, our pipeline consists of three components:

(1) **Environment Gym**: This component simulates events within a specified background setting and example events, providing a sandbox for proactive agents to interact. It has two key functionalities: (i) Event Generation: creating potential sequences of environmental events tailored to specific scenarios; (ii) State Maintenance: updating and maintaining the environment's state when new user activities are generated or when the agent performs actions during task execution.

(2) **Proactive Agent**: This component is responsible for predicting tasks that the user might assign to the agent based on the user's needs as inferred from the event history. It also interacts with tools to complete specific tasks assigned by the user.

(3) **User Agent**: This component simulates the user's activities and responses based on predefined user characteristics. It decides whether to accept and execute the tasks proposed by the agent.

In the following sections, we introduce the details of each component.

## 3.3 ENVIRONMENT GYM

**Event Collection**    To improve the quality of events generated by the environment gym, we first collect real-world events as reference. We developed a monitor software based on Activity Watcher[1], which allows us to capture the details of user interactions with computer systems, including keyboard and mouse operations, visited web pages, and used development tools. To enhance the semantic richness of the collected data and facilitate parsing by large language models, we further merge the raw data into logically coherent segments. Additionally, we utilize a language model to

---

[1] https://github.com/ActivityWatch/activitywatch

translate the structured data into more natural textual descriptions. This process not only improves the interpretability of the data but also makes it more suitable for subsequent usage. Here is a sample event (more samples can be found in Appendix E.1):

```
Time: 1717378968.208
> Event: The user opens a new browser tab and navigates to the Google.
```

**Scenario Generation**   After collecting reference events, rather than directly generating specific events, we generate a realistic interactive scenario to provide sufficient background information first for further generation. To build such scenarios, we first prompt GPT-4o (OpenAI, 2024) with the seed jobs collected from human annotators to create various jobs the user might perform under a specific category, like coding, writing, or daily life. Then, we generate all possible entities that the tasks might involve, e.g. browser, software, and tools for the agent to perform tasks. Next, we refine the scenario by adding more details like entity status or date time to improve the details. Finally, the collected events are also provided to generate example events under each particular context for future events generation. This allows us to control the granularity of events that will be generated and maintain the diversity of the scenarios. See Appendix C for the specific prompt used.

**Event Generation**   When it comes to specific event generation, we start with user activity generation. For each scenario, the user agent is first requested to describe its activities and actions $A_t$ at time $t$ to complete the job in the simulated environment. Then, the gym accepts the user's activities and actions to generate detailed events one by one. As depicted in Figure 2, the gym is tasked to generate logically correct and fluent events according to historical events and the current environmental state. The key to improving the realities of the events generated and adapting to different environments is utilizing the example events we generated based on collected events during scenario generation. Before generating events, we randomly sample the generated example events for the specific scenario and request the gym to produce new events according to them. Once a new event $E_{t+1}$ is generated, the gym updates the entities' status in the environments and repeats the process until there are no events that can be generated with the provided user activities. This comprehensive approach ensures that each subsequent event is not only appropriate but also contributes to a coherent and logical progression within the scenario.

**State Maintenance**   Another important functionality of the gym is maintaining the state of the environment $S_t$. During the scenario generation, the gym creates entities like browsers or development kits in the simulated environment, where each entity has its state and properties like the application version or the specific browser name. When a new event is generated, the gym should update the states and properties of them to provide feedback for further event generation. Specifically, we first gather historical state changes of related entities and prompt the GPT-4o to generate new states of the entities $S_{t+1}$ with the new event. During the process, the simulated time will also be updated according to the granularity of the event. After that, the next event will be generated based on the latest environment state $S_{t+1}$.

### 3.4   PROACTIVE AGENT

The second component in our data generation pipeline is the proactive agent that predicts tasks the user might assign. As detailed in Figure 3, upon the agent receiving new event $E_t$ at time $t$, it first updates its memory with the event. To improve the quality of the prediction, it also accepts feedback from the user agent on its draft prediction. Combining new events with historical ones and conversations with the user, the agent incorporates user characteristics to raise potential tasks. If the agent detects potential tasks, it will raise the task as a new event and wait for the judgment of the user agent. Otherwise, the proactive agent predicts no potential tasks and stays silent. Once the predicted task is accepted, the agent will execute the task within the gym, which generates multiple events about the interaction between the agent and the environment. During the data generation, the agent would constantly receive events from the Gym and predict potential tasks.

**Task Execution**   As mentioned before, the proactive agent executes the predicted task once the user accepts. This process is mainly done through multi-turn interaction between the proactive agent and the gym. Specifically, the proactive agent will be provided with the tools generated during

the scenario generation, like file system tools in the computer or access to the smart light switch, to interact with the simulated environment. Each time the proactive agent takes an action, the gym will generate a new event, which is further processed by the gym and the user agent to update the environment state. After that, the proactive agent detects the new environment state $S_{t+1}$ and takes new actions according to the events generated by the gym. This process ends when the user interrupts or the proactive agent finishes its tasks.

## 3.5 USER AGENT

The user agent is designed to emulate users' activities $A_t$ and responses about the agent's prediction $P_t$, as illustrated in Figure 2. We prompt GPT-4o to generate activities and actions for the provided task in the specific environment. The gym further processes the activities and actions to generate a new event. Then the proactive agent predicts potential tasks according to the events. Upon receiving the predicted task, the user agent determines whether to accept or reject it. If the user agent accepts the task, the proactive agent will set up and execute the accepted task within the environment gym. Otherwise, if the user agent declines the suggested assistance, the environment gym generates new events autonomously without any interventions. In our settings, we collect judgment from human annotators and train a reward model to simulate the judgment.

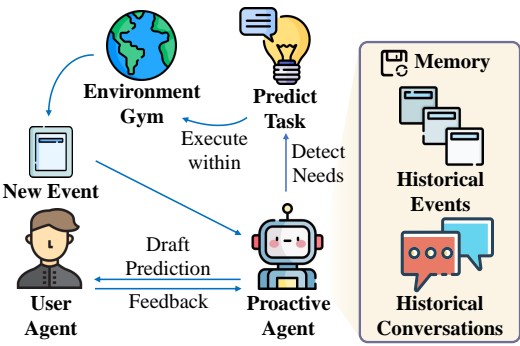

Figure 3: Overview of the proactive agent framework. The agent monitors new events and updates its memory to predict potential tasks.

Specifically, to ensure the reward model aligns closely with human judgment, we generate and annotate a dedicated dataset to indicate whether humans would accept the predicted task or not. We utilize 9 different language models to generate diverse predictions for each event. Between these predictions, We select 5 predictions with the minimum total cosine distance after embedding the predictions using Text-Ada-Embedding to determine our label target. Each prediction is annotated with one of three options by three separate annotators: *accept*, *reject*, or *reject all*. The *reject all* option is chosen when the annotator believes that the given events did not imply any possible tasks that the user might assign, aka $N_t = 0$ in Section 3.1. Otherwise, if one prediction is labeled as accepted, we label the event $E_t$ with $N_t = 1$. We use majority voting to make the final decision on each prediction. After all, the annotation results in a dataset of $1,760$ entries, each containing event traces, task predictions, and decisions on accepting the predicted task from three distinct annotators. Notably, our annotators achieve an impressive agreement ratio of over $91.67\%$ on the test set, underscoring the annotations' reliability and the dataset's robustness for further analysis. To further facilitate automatic data generation, we also prompt the GPT-4o to produce a more detailed explanation of the user judgment. Further details regarding the assessment of the reward model are available in Section 4.1.

## 4 EXPERIMENTS

### 4.1 REWARD MODEL ASSESSMENT

To automatically evaluate whether the predicted tasks and their timing are appropriate, we seek to train a reward model capable of accurately imitating user judgments. To this end, we apply the user-annotated data to train LLaMA-3.1-8B-Instruct (Touvron et al., 2023) and compare it with several baselines to show its superiority.

**Setting.** We use the $1,760$ entries with human annotations and randomly split them into a training set ($1,640$ entries) and a test set ($120$ entries). We then train LLaMA-3.1-8B-Instruct on the training set to obtain our reward model. We employ a total batch size of $32$, a learning rate of $1e - 5$, and

| | GPT-4o | GPT-4o-mini | LLaMA-3.1-8B | LLaMA-3.1-70B | Ours |
|---|---|---|---|---|---|
| Agree. MN$^\uparrow$ | 3.33% | 56.67% | **80.00%** | 33.33% | **80.00%** |
| Agree. NR$^\uparrow$ | **100.00%** | 56.67% | 30.00% | 83.33% | 86.67% |
| Agree. CD$^\uparrow$ | **100.00%** | 86.67% | 96.67% | **100.00%** | **100.00%** |
| Agree. FD$^\uparrow$ | 0.00% | 33.33% | 13.33% | 6.67% | **100.00%** |
| Recall$^\uparrow$ | **100.00%** | 71.67% | 63.33% | 91.67% | 93.33% |
| Precision$^\uparrow$ | 50.42% | 56.58% | 54.29% | 53.40% | **90.32%** |
| Accuracy$^\uparrow$ | 50.83% | 58.33% | 55.00% | 55.83% | **91.67%** |
| F1-Score$^\uparrow$ | 67.04% | 63.24% | 58.46% | 67.48% | **91.80%** |

Table 2: Evaluation results of different models on our test set as the reward model. For the Missed-Need (MN), Non-Response (NR), Correct-Detection (CD), and False-Detection (FD) scenarios, we present the **agreement ratio** between models and the major voting of our human annotators. Our model which is fine-tuned based on LLaMA-3.1-Instruct-8B achieves the best F1-Score of $91.80\%$.

an Adam Optimizer with a $0.1$ warm-up ratio. We train the reward model for $5$ epochs to prevent it from over-fitting. We use 8 A100 GPUs on one node to train for approximately 1.5 hours. The detailed prompt template is listed in Appendix A. We use the test split to evaluate our adapted model and all the baselines. To be noticed, our human annotators achieve up to $91.67\%$ agreement ratio on the test set, demonstrating the effectiveness of our evaluation.

**Metrics.** We use the reward model to perform binary classification on whether to accept predicted tasks and compare its results with human-annotated results. This assesses how well the reward model aligns with human judgment regarding the suitability of the predicted tasks. We compare the judgments made by the reward models and humans to calculate the Recall, Precision, Accuracy, and F1-Score. Additionally, we calculate the agreement ratio for the following cases:

- **Missed-Needed**: $N_t = 1, P_t = \emptyset$, the user needs help, but the agent does not provide it.
- **Non-Response**: $N_t = 0, P_t = \emptyset$, the user does not need help, the agent does not prompt.
- **Correct-Detection**: $N_t = 1, P_t \neq \emptyset$, and the user accepts the task predicted by the agent.
- **False-Detection**: $N_t = 0, P_t \neq \emptyset$, the user does not need help but agent prompts.

**Results.** As Table 2 shows, all existing models perform well on correct detection, but perform badly in other scenarios, especially in the false alarm scenario. After a deeper analysis, we find that existing models just can not infer what the user might need and tend to accept arbitrary help, even if it is very abstract or meaningless to current observation. In contrast, our reward model achieves a $100\%$ agreement ratio on false alarm scenario and a solid $91.80\%$ F1-Score across all scenarios. We select our reward model for further analysis across the ProactiveBench.

## 4.2 PROACTIVE AGENT EVALUATION

**Setting.** We use the training set of ProactiveBench to obtain the Proactive Agent based on the two open-source models: LLaMA-3.1-8B-Instruct and Qwen2-7B-Instruct. During training, we employ a total batch size of $32$, a learning rate of $1e-5$, and an Adam Optimizer with a $0.01$ warm-up ratio. We train the model for $3$ epochs. We use 8 A100 GPUs on one node to train for approximately 2 hours. The detailed prompt can be found in Appendix B. The automatic evaluation of these metrics relies on the simulated judgment given by the reward model. All models are evaluated in our test split of the ProactiveBench, which contains $233$ events collected in the real world. We employ the same prompt template and apply greedy decoding across all models.

**Metrics.** We evaluate the performance of the Proactive Agent based on whether the user accepts its prediction. As described in Section 3.1, the user's acceptance $R_t$ contains four conditions. In our specific settings, **Recall** measures the proportion of actual needs for assistance that were correctly predicted by the agent, including cases where the agent predicts a task and the user accepts it, as well as cases where no task is predicted and the user does not need assistance. **Precision** measures the proportion of predicted tasks that were actually accepted by the user. **Accuracy** measures the overall

correctness of the agent's predictions. The **False-Alarm** measures the proportion of incorrect task predictions, specifically when a task is predicted but not needed. The **F1-Score** provides a balanced measure of the goodness of the agent's proactive behavior. We use the reward model during the evaluation to automatically generate the user's judgment. Based on the confusion matrix, we report Recall, Precision, Accuracy, False Alarm, and F1-Score across all settings. The detailed calculation method can be found in Appendix B.

| Model | Recall$^\uparrow$ | Precision$^\uparrow$ | Accuracy$^\uparrow$ | False-Alarm$^\downarrow$ | F1-Score$^\uparrow$ |
|---|---|---|---|---|---|
| *Proprietary models* | | | | | |
| Claude-3-Sonnet | 27.47% | 37.31% | **52.42%** | 62.69% | 31.65% |
| Claude-3.5-Sonnet | 97.89% | 45.37% | 49.78% | 54.63% | 62.00% |
| GPT-4o-mini | **100.00%** | 35.28% | 36.12% | 64.73% | 52.15% |
| GPT-4o | 98.11% | **48.15%** | 49.78% | **51.85%** | **64.60%** |
| *Open-source models* | | | | | |
| LLaMA-3.1-8B | 98.86% | 38.16% | 39.06% | 61.84% | 55.06% |
| LLaMA-3.1-8B-Proactive | **99.06%** | **49.76%** | **52.86%** | **50.24%** | **66.25%** |
| Qwen2-7B | 98.02% | 44.00% | 43.61% | 56.00% | 60.74% |
| Qwen2-7B-Proactive | **100.00%** | **49.78%** | **50.66%** | **50.22%** | **66.47%** |

Table 3: Evaluation results of different models' performance on the ProactiveBench. The GPT-4o stands out for close-sourced models, achieving over $64.60\%$ F1-Score. For open-sourced models, our fine-tuned Qwen2-7B model achieves best result, with a $66.47\%$ F1-Score.

**Result.** Table 3 compares various models on the test set of the ProactiveBench, which contains 233 events collected from the real world user. Close-sourced models like GPT-4o (OpenAI, 2024) or GPT-4o-mini all tend to predict proactive tasks actively. Most of them succeed in assisting when the user needs but fail to stay silent when the user does not require any assistance, resulting in a relatively high false alarm ratio. For example, the GPT-4o-mini provides unnecessary assistance even when the events provided do not contain meaningful operations, like switching between software but doing nothing else. Another big issue is early assistance when no precise users' intents can be found in the given observation. This makes the proactive tasks proposed by the model seem too abstract or useless, resulting in a relatively high false alarm ratio. The Claude-3-Sonnet (Anthropic, 2024) shows a different example of failing to detect the user's need and provide assistance that does not satisfy the user's expectation.

For open-sourced models, we evaluate the performance of the LLaMA-3.1-Instruct-8B and Qwen2-Instruct-7B before and after fine-tuning based on our synthesized data. As shown in table 3, both models obtain an impressive improvement. For example, the LLaMA-3.1-8B improves its F1-Score from $55.06\%$ to $66.25\%$. The results demonstrate the effectiveness of our data synthesis pipelines. As for the concern of being overly interrupted by the proactive agent, our fine-tuned models achieve solid progress in reducing the false alarm ratio, which is comparable to the performance of the GPT-4o. Besides, the finetuned Qwen2-7B is also outperform the GPT-4o in terms of the F1-Score, resulting in the highest F1-Score of $66.47\%$ However, we also observed the same pattern of models tends to provide as much assistance as possible, instead of providing necessary assistance when the user needs it.

In short, while most models can assist when needed, they still frequently offer unnecessary help, even when instructed to provide only essential assistance.

## 4.3 PERFORMANCE ANALYSIS

In this section, we analyse two possible type of settings that could impact the performance of the proactive agent.

**Predict Multiple Tasks.** When it comes to real-world applications, the proactive agent can provide multiple candidate tasks to improve overall performance. To evaluate how models perform under

| Model | Settings | Recall$^\uparrow$ | Precision$^\uparrow$ | Accuracy$^\uparrow$ | False-Alarm$^\downarrow$ | F1-Score$^\uparrow$ |
|---|---|---|---|---|---|---|
| GPT-4o-mini | pred@1 | 100.00% | 35.28% | 36.12% | 64.73% | 52.15% |
| | pred@3 | 99.32% | 65.32% | 66.52% | 34.68% | 78.80% |
| | w/ RM | 55.45% | 63.54% | 63.95% | 36.46% | 59.22% |
| | pred@3, w/ RM | 100.00% | 65.35% | 66.09% | 34.65% | 79.05% |
| GPT-4o | pred@1 | 98.11% | 48.15% | 49.78% | 51.85% | 64.60% |
| | pred@3 | 100.00% | 63.56% | 64.81% | 36.44% | 77.72% |
| | w/ RM | 56.76% | 55.26% | 57.61% | 44.74% | 56.00% |
| | pred@3, w/ RM | 100.00% | 63.30% | 65.67% | 36.70% | 77.53% |
| LLaMA-3.1-8B | pred@1 | 98.86% | 38.16% | 39.06% | 61.84% | 55.06% |
| | pred@3 | 100.00% | 52.79% | 52.79% | 47.21% | 69.10% |
| | w/ RM | 77.08% | 42.52% | 47.64% | 57.41% | 54.81% |
| | pred@3, w/ RM | 95.12% | 61.58% | 66.09% | 38.42% | 74.76% |

Table 4: Comparison between different settings for each model. The setting "pred@1" means predicting one task at a time. The setting "pred@3" means predicting 3 tasks at a time. The setting "w/ RM" means we will provide feedback from the reward model to help better prediction.

this condition, we allow them to generate multiple candidate tasks at once, but no more than three to avoid a high cognition burden for the user. In this setting, we let the reward model check the candidate tasks one by one. We label the result as accepted if one of the candidate tasks is accepted, and rejected if only all the candidate tasks are rejected.

As shown in Table 4, all models obtain solid improvement across all metrics when comparing "pred@1" with "pred@3". Take the GPT-4o as an example, it obtains higher accuracy and precision while reducing its false alarm by providing diverse candidate tasks. The huge drops in the false alarm ratio, from $51.85\%$ to $36.44\%$ are mainly due to its improvement in providing proactive tasks. However, when comparing GPT-4o-mini with LLaMA-3.1-8B, we observed different degrees of improvement. These two models perform similarly when predicting only one proactive task at once, but show a nearly $9\%$ difference in terms of F1-Score when predicting multiple candidates at once. We analyzed the result and found that the LLaMA-3.1-8B tends to provide unexpected assistance when the user's need is unclear, which can not be improved by providing multiple candidates.

**Feedback From the Reward Model.** We also investigate whether the feedback from our reward model could help models improve their performance on the ProactiveBench. This is done with the same logic as described in Figure 3. For each model, we first ask them to generate a draft prediction and obtain feedback from the user agent (which is built on the reward model in this case). Then we let the model refine its prediction to obtain the final prediction.

As shown in Table 4, by adding the feedback from the reward model (settings with "w/ RM"), models generally reduce their false alarm ratio and improve the accuracy, but drop dramatically in terms of the recall. We observe that models stay silent once they receive feedback from the reward model. However, doing nothing is not always the optimal solution. The GPT-4o seems to fail to capture the possible user needs, leading to a performance drop in terms of F1-Score. For other models like GPT-4o-mini and LLaMA-3.1-8B, they deed obtain a marked improvement in terms of the F1-Score. Another setting combining the multiple-task prediction with the reward model ("pred@3, w/ RM") shows a more general improvement across the board. By integrating the reward model into the Proactive Agent, we can make the Proactive Agent more smartly detect user needs and reduce the false alarm ratio even when we can not access the weight directly, which is good news for developing the Proactive Agent.

## 4.4 CASE STUDY

In this section, we explore two prevalent types of failures encountered in predicting possible tasks: the inability to detect user needs and making predictions at inappropriate times. More detailed failure modes can be found in Appendix D.

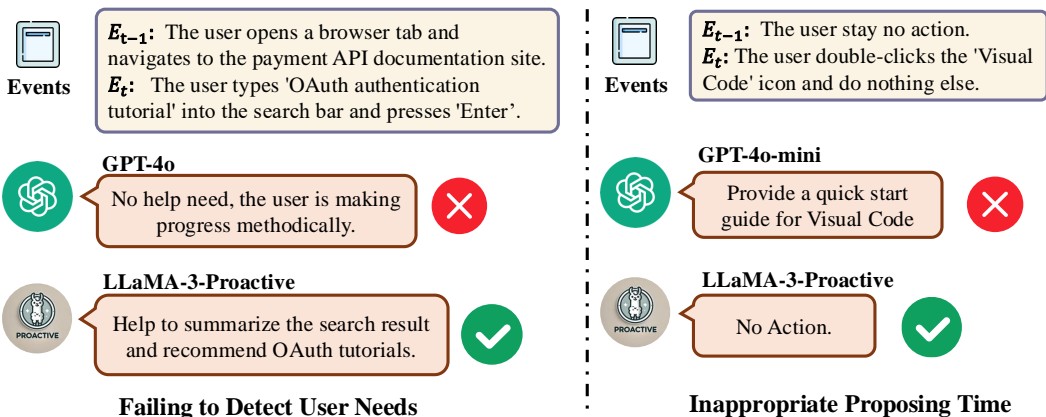

Figure 4: Two types of failure: failing to detect user needs (left) and inappropriate proposing time (right). We compare the response between our fine-tuned LLaMA-3.1-Instruct-8B with other models to show the refined proactive behavior of the model.

As illustrated in Figure 4 (left), a notable failure occurs when the GPT-4o model does not assist at crucial moments. For instance, when a user is engaged in integrating a payment API and requires a tutorial for guidance, the model remains silent. Instead, our model successfully detects human needs and offers assistance. The underlying intention to minimize disruptions ironically leads to missed opportunities to offer timely help.

Conversely, the right side of Figure 4 showcases an instance of ill-timed prediction. Here, the GPT-4o-mini suggests an action when there are no user needs showing in events. This scenario underscores the possible unintended events existing in human activities. The model should judge whether there are possible tasks smartly to avoid unnecessary actions. These instances highlight the intricate nature of human activities and the sophisticated reasoning required for models to accurately predict human needs. To navigate the delicate balance between being helpful and intrusive, models must develop a deeper understanding of user contexts and activities, ensuring their interventions are both timely and pertinent.

## 5 CONCLUSION

We present an innovative approach to human-agent interaction by leveraging proactive task predictions that anticipate human needs. We introduce ProactiveBench, a comprehensive dataset comprising 6,790 events, designed to refine the proactive behavior of LLM-based agents and establish an automatic benchmark for assessing model proactiveness. By iteratively generating events in synthesized scenarios, we create training data that enhances the proactive capabilities of our models. Our experiments demonstrate significant improvements in the agent's performance on ProactiveBench, validating the effectiveness of our methods. Despite these advancements, our findings underscore ongoing challenges, particularly in minimizing inappropriate task proposals and ensuring task predictions are contextually accurate. Future research should focus on enhancing the precision and timeliness of task predictions to improve the efficacy of the proactive human-agent interaction.

## ETHICS STATEMENT

In the construction of our dataset, we have adhered to strict ethical standards. The human annotators are students from the computer sciences area and obtain rewards for their annotations. We use ChatGPT in this research. All the original data utilized for the compilation of our dataset are either sourced from open-access platforms or have been used with the appropriate permissions granted. We have conducted a thorough review to ensure that our dataset is devoid of any harmful content or private information. Our commitment to ethical research practices extends to the respect for privacy, intellectual property rights, and the integrity of the data sources.

ACKNOWLEDGMENT

This work was supported in part by the Major Key Project of PCL (Grants PCL2024A06 and PCL2022A05), the Shenzhen Science and Technology Program (Grant RCJC202331211085918010), the Postdoctoral Fellowship Program of CPSF (Grant GZB20230343), and the China Postdoctoral Science Foundation (Grant 2023M741945).

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

APPENDIX

LIMITATIONS

While our method demonstrates that it can effectively and proactively predict possible tasks, the current research is constrained by several limitations. Firstly, the environment settings we have explored are still limited. The contexts in this paper provide a foundational understanding, but broader application areas need to be investigated to fully establish the versatility and robustness of the proactive agent. Moreover, models still exhibit a relatively high ratio of false alarms, indicating that they cannot yet perfectly predict possible tasks. This limitation highlights the need for further refinement of the model's proactive behavior to avoid bothering the user. The high rate of false positives can lead to unnecessary or incorrect actions, which may reduce user trust and the overall efficiency of the system. Future research should focus on several key areas to address these limitations:

- **Expansion of Environment Settings:** Research should explore a wider variety of scenarios and contexts to validate the model's generalizability. This includes domains where proactive prediction of tasks can significantly enhance user experience and operational efficiency.
- **Improvement in Prediction Accuracy:** Efforts should be directed towards reducing the false alarm rate by enhancing the model's understanding of context and user behavior.
- **User-Centric Evaluation:** Future studies should involve extensive user-centric evaluations to better understand how users interact with the proactive agent and to identify areas for improvement. User feedback and behavioral data can provide valuable insights into refining the prediction algorithms and making the system more intuitive and reliable.
- **Ethical and Privacy Considerations:** As the proactive agent needs the environment information for prediction tasks, it is crucial to address ethical and privacy concerns. Ensuring that user data is handled responsibly and that the agent operates transparently and within ethical guidelines will be critical for gaining user trust and acceptance.

## A  REWARD MODEL TRAINING SETTING

We use Llama-3.1-Instruct-8B as the base model for our training. The total dataset size is approximately $1,640$. Specifically, we employ a total batch size of $32$, a learning rate of $1e-5$, and an Adam Optimizer with a $0.1$ warm-up ratio. We train the reward model for 5 epochs to prevent it from over-fitting. We use 8 A100 GPUs on one node to train for approximately 1.5 hours.

*Prompt Template*

```
<Task>
Evaluate the task proposed by the proactive assistant as the user.
</Task>

<Rule>
0. Analyze the current observation to understand your current
   situation and requirements.
1. If the proposed task is 'null' (indicating no task is proposed
   under the current observation), follow these steps:
   - Accept the 'null' task if you believe there is no need for a
     task.
   - Reject the 'null' task if you believe a task is needed.
2. Minimize interruptions from the assistant by only accepting
   tasks that are valuable.
3. Evaluate the current observation and make a judgment on the
   proposed task accordingly.
</Rule>

<Format>
You should answer with the following JSON format:
{
```

```
    "thought": "Give your thoughts first, then provide the
        judgment of the task.",
    "judgment": "accepted or rejected"
}
</Format>
```

## B AGENT MODEL TRAINING SETTING

Similarly, we use Llama-3-Instruct 8B and Qwen2-Instruct-7B as the base model for agent model training. The total dataset size is approximately $6,790$. Specifically, we employ a total batch size of 32, a learning rate of $1e - 5$, and an Adam Optimizer with a $0.1$ warm-up ratio. We train the model for 3 epochs to prevent it from over-fitting. We use 8 A100 GPUs on one node to train for approximately 2 hours.

**Template.** We apply the following prompt template to train the agent model:

---

*Prompt Template*

```
<Role> You are a helpful assistant that provides proactive
    suggestions to the user. </Role>

<Task> Understand what the user is doing and anticipate their
    needs based on events. Only propose assistance when you fully
    understand the user's actions. Use available operations to
    ensure the task is feasible. Execute the task if the user
    accepts your proposal. </Task>

<Format> Respond in the following JSON format:
{
    "Purpose": "The purpose of the user's last action.",
    "Thoughts": "Your thoughts on the user's actions.",
    "Proactive Task": "Describe your proposed task, or set to '
        null' if no assistance is needed.",
    "Response": "Inform the user about your assistance if
        proposing a task."
}
</Format>

<Rules>
- Ensure the proposed task is relevant to the events. - Focus on
    the user's current needs and predict helpful tasks.
- Consider the timing of events.
- Only offer proactive assistance when necessary.
- Deduce the user's purpose and whether they need help based on
    event history.
- Set 'Proactive Task' to 'null' if the user doesn't need help.
</Rules>
```

## C PROMPT TEMPLATE FOR ENVIRONMENT GYM

### C.1 PROMPT FOR SCENE GENERATION

---

*Prompt Template*

```
<Role>
```

```
You are tasked with simulating an environment within a system. The
    content labeled 'Source: environment' reflects your past
    actions and decisions.
</Role>

<Task>
Generate and refine detailed environment settings. Based on the
    latest activities, create multiple events to describe changes
    in the environment.
</Task>

<Rules>
- Ensure the subject of the generated content aligns with the
    latest activities's source.
- Avoid subjective opinions or emotions; focus on objective
    changes.
- Ensure events are consistent with historical events labeled '[
    events]' and include all - changes from the activities.
- Introduce occasional failures or unexpected events for realism.
- Ensure each event is logically connected to the previous one and
    does not include nonexistent elements.
- Pay close attention to entity operations; if an operation is not
    allowed or impractical in the real or simulated environment,
    raise an error and explain the issue.
</Rules>
```

## C.2 SEED JOBS DATA

*Prompt Template*

```
<Task>
You are tasked to generate realistic scenarios where a user might
    need assistance from an AI assistant. Always remember to keep
    the scene realistic and believable by including as much
    details as possible.
</Task>

<Rule>
- You will iteratively generate more information about the scene.
    Make sure each time you add a new detail, it is consistent
    with the previous details. Always generate new content based
    on the previous generated content.
- You can add as many details as you want, but make sure they are
    consistent with the previous details.
- Try to generate diverse details about the scene. You will be
    tasked to simulate events in the scene later.
</Rule>
```

## C.3 PROMPT FOR USER AGENT GENERATION

*Prompt Template*

```
<Role>
You are tasked with simulating a user within a system. The content
    labeled 'Source: user' reflects your past actions and
    decisions.
</Role>
```

```
<Task>
Generate human-like activities with distinct characteristics and
    identities. You will receive events and observations from the
    environment; analyze these closely to decide your actions.
</Task>

<Rules>
- Respond like a real user; don't be overly predictable.
- Refer to # User Info to understand your identity.
- Critically evaluate the received information, as it may not
    always be accurate.
- Stay aware of environmental changes, which can occur at any time
    .
</Rules>
```

## C.4 PROMPT FOR STATUS UPDATING

*Prompt Template*

```
<Task>
Evaluate the task proposed by the proactive assistant as the user.
</Task>

<Rule>
0. Analyze the current observation to understand your current
    situation and requirements.
1. If the proposed task is 'null' (indicating no task is proposed
    under the current observation), follow these steps:
    - Accept the 'null' task if you believe there is no need for a
        task.
    - Reject the 'null' task if you believe a task is needed.
2. Minimize interruptions from the assistant by only accepting
    tasks that are valuable.
3. Evaluate the current observation and make a judgment on the
    proposed task accordingly.
</Rule>

<Format>
You should answer with following JSON format:
{
    "thought": "Give your thoughts first, then provide the
        judgement of the task.",
    "judgement": "accepted or rejected"
}
</Format>
```

## C.5 METRICS CALCULATION

**Definition** Here is how we define the label of each prediction.

• **True Positive (TP):** Agent predicts task, the user accepts.
• **False Positive (FP):** Agent predicts task, the user rejects.
• **True Negative (TN):** Agent does not predict a task, and the user does not need assistance.
• **False Negative (FN):** Agent does not predict the task, but the user needs assistance ($N_t = 1$ in Section 3.1).

**Recall**  A high recall indicates that the agent frequently identifies situations where help is needed. This metric is crucial for assessing the agent's ability to recognize and respond to user needs on time.

$$Recall = \frac{TP}{TP + FN} \tag{4}$$

**Precision**  A high precision indicates that the agent proposes good tasks while not bothering the user too much. This metric is crucial when considering the annoying behavior of the proactive agent could greatly reduce user satisfaction.

$$Precision = \frac{TP}{TP + FP} \tag{5}$$

**Accuracy**  High Accuracy demonstrates that the agent has a good understanding of user needs, as most of its predictions are accepted. This metric is essential for measuring the relevance and correctness of the agent's proactive suggestions.

$$Accuracy = \frac{TP + TN}{P + N} \tag{6}$$

**F1-Score**  High F1-Score means the proactive agent strikes a good balance between being helpful and proactive.

$$F_1 = 2 * \frac{Recall * Precision}{Recall + Precision} \tag{7}$$

## D  FAILURE MODES ANALYSIS

---

*Not A Task*: The model does not give a real task at all.

```
Observation: {
    "time": "1717335907.603",
    "event": "The user switches to Visual Studio Code."
}
```

```
GPT-4o-mini Response: It seems you may be switching to Visual
    Studio Code for development. If you need any coding tips,
    resources, or help with Swift UI in VS Code, feel free to ask!
```

Under this event, the agent did not take the event as an action, but rather a call for chat. This chat is not a task we want.

---

*Unspecific Task*: The proposed task is non-executable due to a lack of a clear and detailed description.

```
Observation: {
    "time": "1717329066.416",
    "event": "The user performed a search for 'js fade in/out' on
        Bing."
}
```

```
GPT-4o-mini Response: Compile a list of JavaScript resources,
    tutorials, and documentation that cover advanced topics
    related to the user's search.
```

Under this event, the model proposal is not clear (not pointing out the user's research content) and impractical (Compiling a list of resources, tutorials, and documentation, which is too much for the user and also out of the capability of the model).

---

*Irrelevant Task*: The proposed task is not related to the current context or meets the user's needs.

```
Observation: {
    "time": "1717336090.982",
    "event": "The user clicks, types 'list.swift' in Visual Studio
        Code."
}

GPT-4o Response: I can help by providing tips on how to
    efficiently navigate and manage files in Visual Studio Code or
    offer assistance with the content of 'list.swift'.
```

Under this event, the user created a new file in vscode. However, the agent provided a task about file management, which is irrelevant to the user's intention.

---

*Unwanted Proposal*: The user does not want to be interrupted.

```
Observation: {
    "time": "1717329211.15",
    "event": "The user opened 'http://127.0.0.1:5500/main.html' in
        the web browser."
}

GPT-4o Response: Offer to help troubleshoot or verify the 'fade in
    fade out' effect if it doesn't work as expected.
```

Under this context, since whether the edition is successful or not is not clear, the agent should remain silent rather than propose troubleshooting currently.

## E  DATA EXAMPLES

### E.1  EVENT SAMPLES

---

*Collected Raw Data*

```
[{
    "timestamp": 1717335890.127,
    "duration": 2.056,
    "user_input": [],
    "status": "not-afk",
    "app": "web",
    "events": []
},
{
    "timestamp": 1717335893.215,
    "duration": 10.267,
    "user_input": [
        {
            "from": "mouse",
            "data": {
                "type": "click",
                "button": "left"
            }
        },
        {
            "from": "keyboard",
            "type": "input",
            "data": "swift ui ctrl_l liebiao "
        },
        {
```

```
            "from": "keyboard",
            "data": {
                "type": "pressAndRelease",
                "key": "enter"
            }
        }
    ],
    "status": "not-afk",
    "app": "web",
    "events": []
},
{
    "timestamp": 1717335904.513,
    "duration": 0.0,
    "user_input": [],
    "status": "not-afk",
    "app": "web",
    "events": []
}]
```

*Processed Event*

```
[{
    "time": "1717378968.208",
    "event": "The user opens a new browser tab and navigates to
        the Google homepage."
},
{
    "time": "1717378971.255",
    "event": "The user switches to the 'Code.exe' application but
        does not perform any specific actions."
},
{
    "time": "1717378975.29",
    "event": "The user continues to remain on the 'Code.exe'
        application without performing any actions."
}]
```

*Annotated Data for Reward Model*

```
{
    "obs": [
        {
            "time": "1717338232.283",
            "event": "The user types 'badrandResearch.m' in Visual
                Studio Code."
        },
        {
            "time": "1717338245.419",
            "event": "The user types a Markdown entry about
                research on sustainable branding in Visual Studio
                Code."
        },
        {
            "time": "1717338245.419",
            "event": "The user works on the file 'brandResearch.md
                ' in the project 'SustainableBrandResearch' in
                Visual Studio Code."
```

```
        },
        //...
        {
            "time": "1717338752.214",
            "event": "The user interacts with Katla's website,
                viewing eco-friendly loungewear products made by
                the sea."
        },
        {
            "time": "1717338804.867",
            "event": "The user writes Markdown notes in Visual
                Studio Code about Katla's product pricing."
        }
    ],
    "pred_task": null,
    "valid": false,
    "help_needed": true,
    "annotation": [
        false,
        false,
        false
    ],
    "category": "Missed-Need (MN)"
}
```

## E.2 PROACTIVE AGENT'S PREDICTION

*Testing Data Sample*

```
{System Prompt For Proactive Agent}

User:
{
    "Instructions": "Now analyze the history events and provide a
        task if you think the user needs your help.",
    "Observation": [
        // ...
        {
            "time": "1717337616.208",
            "event": "The user switched back to Visual Studio Code
                 and typed insights related to digital payment
                 systems improving public transportation and
                 reducing the need for private car usage."
        },
        {
            "time": "1717337282.329",
            "event": "The user moved the mouse and clicked the
                left button within Visual Studio Code."
        }
    ]
}

Possible Agent Response:
{
    "Purpose": "The user is currently interacting with a web
        application.",
    "Thoughts": "Based on the absence of a specific action, the
        user might be navigating or searching for content.",
```

```
    "Proactive Task": null,
    "Response": "I see you're actively using the web application.
        If you need help finding something specific or have any
        questions, feel free to ask!"
}
```

---

*Example 1*

```
{
    "observation": {
        "time": "1717378968.208",
        "event": "The user opens a new browser tab and navigates
            to the Google homepage."
    },
    "agent_response": [
        "Suggest checking the user's search history and providing
            personalized search recommendations."
    ],
    "task_status": false,
    "other_infomation": {
        "Purpose": "The user is opening a new browser tab and
            navigating to the Google homepage.",
        "Thoughts": "Based on the event, it seems the user is
            starting a new search or browsing session. They might
            need help with organizing their search or finding
            relevant information.",
        "Response": "Would you like me to review your search
            history and provide some personalized recommendations
            to help you find what you're looking for?"
    },
    "judgment": [
        false
    ]
}
```

---

*Example 2*

```
{
    "observation": {
        "time": "1717378971.255",
        "event": "The user switches to the 'Code.exe' application
            but does not perform any specific actions."
    },
    "agent_response": [
        "Suggest creating a new terminal or command prompt within
            the 'Code.exe' application to help the user with
            coding tasks."
    ],
    "task_status": false,
    "other_infomation": {
        "Purpose": "The user is switching to the 'Code.exe'
            application but didn't perform any specific actions.",
        "Thoughts": "Given the user's recent switch to the 'Code.
            exe' application, it's likely they're preparing to
            start coding or editing. However, they might need help
             with organizing their code, finding relevant
            documentation, or setting up their environment.",
```

```
        "Response": "I can help you create a new terminal or
            command prompt within the 'Code.exe' application. This
             will allow you to easily access coding tools and
            commands. Would you like me to set that up for you?"
    },
    "judgment": [
        false
    ]
}
```

*Example 3*

```
{
    "observation": {
        "time": "1717378975.29",
        "event": "The user continues to remain on the 'Code.exe'
            application without performing any actions."
    },
    "agent_response": [
        "Offer to provide coding assistance, such as suggesting
            online resources, tutorials, or even helping with
            writing code."
    ],
    "task_status": false,
    "other_infomation": {
        "Purpose": "The user is remaining on the 'Code.exe'
            application without performing any actions.",
        "Thoughts": "Given the user's recent actions, it seems
            they're hesitant to start coding or editing. They
            might be experiencing writer's block, struggling with
            a particular concept, or needing help with a specific
            coding task.",
        "Response": "If you're struggling with a coding task or
            need help with a specific concept, I'm here to assist.
             Would you like me to suggest some online resources,
            tutorials, or even help with writing code together?"
    },
    "judgment": [
        false
    ]
}
```

### E.3    REWARD MODEL JUDGMENT EXAMPLE

*Input Prompt*

```
{
    "Observations (Time Ascending)": [
        {
            "time": "1717342908.098",
            "event": "The user searched for 'remote working
                software' in the web browser and pressed 'Enter'."
        },
        {
            "time": "1717342914.314",
            "event": "A new tab titled 'new Tab' was opened in the
                 web browser."
```

```
        },
        {
            "time": "1717342940.516",
            "event": "The user opened a search result with the
                title 'remote working software - search' on Bing."
        },
        {
            "time": "1717342956.012",
            "event": "The user switched to another tab in the web
                browser, interacting with multiple scroll actions
                ."
        },
        {
            "time": "1717343061.447",
            "event": "The user resumed browsing on Bing search."
        },
        {
            "time": "1717343082.081",
            "event": "The user continued exploring search result
                pages in Bing with multiple scrolling actions and
                clicking on specific results."
        }
    ],
    "Proposed Task": null,
    "Instruction": "Now give your judgment. You should complete
        the reasoning process in the first person."
}
```

---

*Ouput Judgment*

```
{
    "thought": "I have been browsing the web and switching between
        different applications, and I haven't received any task
        proposal from the proactive assistant. Upon analyzing my
        current activities, I realize I've been searching for
        remote working software and exploring different search
        results on Bing. Given my current engagement and the fact
        that I haven't been interrupted or indicated a need for
        assistance, I believe there is no need for a task at this
        moment. The assistant's decision to not propose a task
        seems appropriate, as introducing a new task could
        potentially disrupt my workflow and cause unnecessary
        interruptions.",
    "judgment": "accepted"
}
```

## F  ANNOTATION GUIDELINES

---

*Annotation Guidelines*

```
In this annotation task, you need to observe two elements:
The sequence of events is described in the Observations module,
    where each event has an event field detailing what occurred.

Tasks generated by the model in the Tasks module are related to
    the current event sequence.
```

You need to carefully review the five tasks provided by the model.
    Analyze which tasks align with your potential needs in the
    context of the described event sequence, without excessively
    disrupting your normal workflow. Then, select the
    corresponding tasks in the checkboxes below. For example, if
    you believe that in the above example, tasks 3 and 4 are what
    you expect the agent to perform, and the others are not, then
    check the boxes for tasks 3 and 4.

If you think none of the tasks are appropriate (or you do not want
    to be disturbed by the agent in the current situation),
    please select the reject all checkbox.

After making your selection, click the continue button to submit
    the annotation results. At this point, new events will be
    updated in the observations, and you will repeat the above
    process until the entire event sequence is annotated.

### Special Events
In addition to regular task annotations, some scenarios offer two
    choices, requiring you to determine whether help is needed
    given the observation. In this case, you need to choose
    between task1 and task2 (note: do not select reject all).

