# OpenReview forum: "Proactive Agent: Shifting LLM Agents from Reactive Responses to Active Assistance"
_ICLR.cc/2025/Conference — ICLR 2025 Poster_

### Official Review · Reviewer_fw7i · 2024-10-16

**Soundness:** 2
**Presentation:** 3
**Contribution:** 2
**Rating:** 5
**Confidence:** 4

**Summary:**

This paper develops proactive agents—agents capable of initiating tasks without explicit human instructions by simply observing the environmental state and historical actions. This direction is important as it represents a shift from reactive to proactive agents, a very useful characteristic for future agents.

To train such a proactive agent, the authors propose a data synthesis framework. They utilize GPT-4o to construct a text-based simulation environment called Environment Gym. This 'gym' functions as a sandbox that employs GPT-4o to generate new environmental states whenever agents perform actions. Additionally, the authors prompt GPT-4o to predict proactive actions through a proactive agent module and simulate user interactions to determine whether to accept the predicted actions.

Through this data pipeline, combined with some minor human annotations, the authors curated 1,760 data entries, using 120 of them for testing. The trained proactive agent demonstrates a clear performance improvement over the vanilla LLaMA-3.1-8B model and GPT-4o, which, while notable, is not entirely surprising.

**Strengths:**

1.  The concept of creating proactive agents is important yet less explored in the field. This paper represents a significant early step in that direction. While proactive conversational models are not entirely new, applying this approach to the realm of agents should be regarded as a noteworthy contribution to the field.

2. The curated benchmark and the proposed proactive agent prompts will serve as valuable resources for the community.

**Weaknesses:**

One major weakness I identified is the text-based simulation environments. In the current implementation, all environmental changes and updates are managed by GPT-4o. This raises concerns about whether LLMs can accurately reflect true environmental changes. Additionally, this reliance on LLMs could potentially make the benchmark too easy, as they may only be capable of predicting changes that they fully understand and are familiar with, which are also easily solvable by other LLMs.

After reviewing the entire paper and appendix, I still find myself unclear about the specifics of the dataset. Including some statistics and examples in the main text would be very helpful.

As stated in the introduction, the main contributions of this paper are the data pipeline and the proactive agent. However, the method used for creating this agent is just simple fine-tuning. Without comparison with strong baselines, it's really hard to assess the true effectiveness of the proposed data pipeline.  I understand the difficulty of finding baselines for new problems, but the current evaluation results are not very convincing.

**Questions:**

Would it be possible to turn text-based simulation into a real interactive environment?

Can you propose more baselines or analyses to demonstrate the effectiveness of the proposed agent?

---

> ### Author Response · Authors · 2024-11-22
> **Statics and examples of the dataset.**
>
> # W2. Statics and examples of the dataset.
> Your summary mentioned that:
> >Through this data pipeline, combined with some minor human annotations, the authors curated 1,760 data entries, using 120 of them for testing.
>
> Here, we would also like to clarify some misunderstandings regarding our dataset and the extent of human annotations involved.
> Our dataset comprises four parts:
> 1. Dataset for the Agent Model
>   - Train set: 6,790 entries (Synthetic)
>   - Test set: 233 entries (Human-Collected)
> 2. Dataset for the Reward Model
>   - Train set: 1,640 entries (Human-Annotated)
>   - Test set: 120 entries (Human-Annotated)
>
> It's important to note that both the test set for the agent model and the entire dataset for the reward model are sourced from real-world data and meticulously annotated by our human annotators. This process resulted in nearly 2,000 human-annotated entries.
>
> Furthermore, each entry in the reward model dataset is annotated at least three times, totaling nearly 6,000 annotation labels. This level of annotation effort is substantial and goes beyond what might be considered "minor human annotations".
>
> We hope this clarification accurately reflects the scope of our work and addresses your concerns. **We‘ve also updated Table 1 in our paper to better clarify this.**
>
> :) Thank you for understanding.
>
> Here are more examples from our dataset that can help you understand our dataset (we've also updated the appendix):
>
> ```text
> Events Data
> -----------
> {"time": "1717377997.0","event": "The user starts working on a file 'CheckName.rb' within a Ruby project in Visual Studio Code."}
> {"time": "1717342826.292","event": "The user started browsing the web."}
> {"time": "1717335993.089","event": "The user scrolls multiple times on a web page."}
>
> Annotated Data for Reward Model
> -------------------------------
> {
>     "obs": [
>         {
>             "time": "1717338232.283",
>             "event": "The user types 'badrandResearch.m' in Visual Studio Code."
>         },
>         {
>             "time": "1717338245.419",
>             "event": "The user types a Markdown entry about research on sustainable branding in Visual Studio Code."
>         },
>         {
>             "time": "1717338245.419",
>             "event": "The user works on the file 'brandResearch.md' in the project 'SustainableBrandResearch' in Visual Studio Code."
>         },
>         //...
>         {
>             "time": "1717338752.214",
>             "event": "The user interacts with Katla's website, viewing eco-friendly loungewear products made by the sea."
>         },
>         {
>             "time": "1717338804.867",
>             "event": "The user writes Markdown notes in Visual Studio Code about Katla's product pricing."
>         }
>     ],
>     "pred_task": null,
>     "valid": false,
>     "help_needed": true,
>     "annotation": [
>         false,
>         false,
>         false
>     ],
>     "category": "Missed-Need (MN)"
> }
>
>
> Training Data for Reward Model
> ------------------------------
> {System prompt for the reward model, please refer to the appendix.}
>
> User:
> {
>     "Observations (Time Ascending)": [
>         {
>             "time": "1717337616.208",
>             "event": "The user switched back to Visual Studio Code and typed insights related to digital payment systems improving public transportation and reducing the need for private car usage."
>         },
>         {
>             "time": "1717337282.329",
>             "event": "The user moved the mouse and clicked the left button within Visual Studio Code."
>         }
>     ],
>     "Proposed Task": "Provide additional research articles on digital payment systems and their impact on traffic management",
>     "Instruction": "Now give your judgment. You should complete the reasoning process in the first person."
> }
>
> Assistant:
> {
>     "thought": "I have already conducted research on digital payment systems and their impact on traffic management. I have also taken notes and compiled relevant insights in my markdown file. The proposed task seems redundant as I have already made progress on this topic. I have found relevant information through Google search results and have reflected on the strategies to improve public traffic, including digital payment systems. Therefore, I do not see the need for additional research articles on this topic.",
>     "judgment": "rejected"
> }
>
> Test Data for Agent Model
> -------------------------
> {System prompt for the agent model, please refer to the appendix.}
>
> User:
> {
>     "Instructions": "Now analyze the history events and provide a task if you think the user needs your help.",
>     "Observation": "[placeholder]"
> }
>
> Possible Agent Response:
> {
>     "Purpose": "The user is currently interacting with a web application.",
>     "Thoughts": "Based on the absence of a specific action, the user might be navigating or searching for content.",
>     "Proactive Task": null,
>     "Response": "I see you're actively using the web application. If you need help finding something specific or have any questions, feel free to ask!"
> }
> ```

---

> ### Author Response · Authors · 2024-11-22
> **Clarification of evolving environmental states with GPT models; Turn text-based simulation into a real interactive environment.**
>
> # W1. Clarification of evolving environmental states with GPT models.
>
> The primary goal of integrating evolving environmental states with GPT models is to generate more coherent and logically flowing events, rather than achieving precise state change modeling. These states act as contextual reminders within our environmental simulations, aiding the model in crafting new events. The core purpose of these states is to introduce variability in the events generated by the model, enabling it to anticipate "what might happen next" or consider "what unusual events could occur."
>
> One of the main challenges in our approach is not the model's inability to accurately simulate an environment, but rather its tendency to create overly perfect or neutral scenarios. The real world is in chaos. Users often perform seemingly random actions, such as clicking aimlessly, moving the mouse erratically, or hesitating over words they can't spell, leading to a plethora of confusing actions. Current models struggle to replicate this unpredictability.
>
> By incorporating detailed states and aligning generated events with real-world occurrences, we've found that our simulation environment can introduce a degree of uncertainty and reality. For instance:
>
> ```text
> {"time": "10-18 09:17:40", "content": "User inputs 'ctrl_l' with mouse interactions, clicks in an application."}
> {"time": "09-17 15:46:53", "content": "The user double-checks the warranty information for the shortlisted devices before making a purchase."}
> {"time": "10-05 14:26:56", "content": "The user uses 'ctrl+c' to copy data about material sustainability from EcoChic\u2019s website for documentation."}
> ```
> The final performance gain also demonstrates the effectiveness of our method.
>
> Besides, you mentioned that:
>
> > Additionally, this reliance on LLMs could potentially make the benchmark too easy, as they may only be capable of predicting changes that they fully understand and are familiar with, which are also easily solvable by other LLMs.
>
> We suggest that this concern comes from a crucial misunderstanding of our dataset. As the test set is totally collected from real-world annotators, they are not easily solvable by LLMs.
>
> # Q1. Turn text-based simulation into a real interactive environment.
>
> As described in W1, when comes to generating data to train the agent model, we suggest that the major problem is to simulate user activities rather than collect real environment states.
> Turning text-based simulation into a real interactive environment can not improve the uncertainty or randomness of simulated user activities. However, it will indeed complicate the setup of diverse application scenarios and greatly slow down the data generation.
> We also argue that this can improve the reality of events generated and might be a good idea in certain situations.
>
> # W3 Q2. More baselines to demonstrate the effectiveness of the data pipelines.
> We've extended our baseline settings to remove the Chain-of-Thought prompt. We also present more experiments about enhancing the proactiveness of agents with the reward model.
> Here are added experiments:
> | Model        | Settings  | Recall  | Precision | Accuracy | False-Alarm | F1-Score  |
> |--------------|-----------|---------|-----------|----------|-------------|-----------|
> | GPT-4o-mini  | w/o CoT   | 100.00% | 18.83%    | 22.32%   | 81.17%      | 31.70%    |
> |              | w/  CoT   | 100.00% | 35.28%    | 36.12%   | 64.73%      | 52.15%    |
> | GPT-4o       | w/o CoT   | 88.06%  | 39.33%    | 57.51%   | 60.67%      | 54.38%    |
> |              | w/  CoT   | 98.11%  | 48.15%    | 49.78%   | 51.85%      | 64.60%    |
> | LLaMA-3.1-8B | w/o CoT   | 100.00% | 36.52%    | 37.34%   | 63.48%      | 53.50%    |
> |              | w/  CoT   | 98.86%  | 38.16%    | 39.06%   | 61.84%      | 55.06%    |
> |              | finetuned | 99.06%  | 49.76%    | 52.86%   | 50.24%      | 66.25%    |
> | Qwen2-7B     | w/o CoT   | 98.80%  | 36.77%    | 39.06%   | 63.23%      | 53.59%    |
> |              | w/  CoT   | 98.02%  | 44.00%    | 43.61%   | 56.00%      | 60.74%    |
> |              | finetuned | 100.00% | 49.78%    | 50.66%   | 50.22%      | 66.47%    |
>
> Hopes this can solve your concerns.
> If you have any additional questions or require further clarification, please feel free to reach out.

---

> > ### Comment · Reviewer_fw7i · 2024-11-25
> >
> > Thank you for the clarification on the dataset details. Although I can relate that simulate user activities is important, I don't agree that collecting real env states are not important.
> >
> > Regarding the added results, can you explain why we observe a high recall?

---

> > > ### Author Response · Authors · 2024-11-25
> > > **Collecting Real Environment States and High Recall Explaination**
> > >
> > > Thank you for your feedback and for highlighting these important points.
> > >
> > > # Q1: Collecting Real Environmental States
> > >
> > > Firstly, our results demonstrate that generating environmental states from models can enhance model performance. This is an important first step, indicating that a model's proactiveness can be improved through a data flywheel approach, even with synthesized data.
> > >
> > > Secondly, we have already incorporated some real environmental states in the events collection part. For example:
> > > ```text
> > > {
> > >     "timestamp": 1717343098.667,
> > >     "duration": 7.229,
> > >     "user_input": [
> > >         {
> > >             "from": "keyboard",
> > >             "type": "input",
> > >             "data": "NC\n"
> > >         },
> > >         {
> > >             "from": "mouse",
> > >             "type": "scroll",
> > >             "data": {
> > >                 "x": 1592,
> > >                 "y": 1591,
> > >                 "scrollx": 0,
> > >                 "scrolly": -2
> > >             }
> > >         },
> > >         {
> > >             "from": "mouse",
> > >             "type": "scroll",
> > >             "data": {
> > >                 "x": 1626,
> > >                 "y": 1513,
> > >                 "scrollx": 0,
> > >                 "scrolly": -1
> > >             }
> > >         }
> > >     ],
> > >     "status": "not-afk",
> > >     "app": "other",
> > >     "events": []
> > > }
> > >
> > > {
> > >     "status": "not-afk",
> > >     "app": "web",
> > >     "events": [
> > >         {
> > >             "url": "edge://newtab/",
> > >             "title": "",
> > >             "audible": false,
> > >             "incognito": false,
> > >             "tabCount": 5
> > >         },
> > >         {
> > >             "url": "https://bing.com/search?pglt=161&q=splashtop&cvid=38bff6321b344b8f9cd074...",
> > >             "title": "splashtop",
> > >             "audible": false,
> > >             "incognito": false,
> > >             "tabCount": 5,
> > >             "innerText": "Splashtop\nhttps://www.splashtop.com/downloads\nSoftware\n|\nBusiness\nSplashtop...splashtop.com\nMicrosoft Store splashtop.cn\nhttps://www.splashtop.cn/cn/remote-desktop-.."
> > >         }
> > >     ]
> > > }
> > > ```
> > > To integrate these real collected environment states into data generation, we organize them into events that are more interpretable for language models. We then sample these events to help the gym generate better events.
> > >
> > > We adopt this indirect approach because modeling realistic interactions between simulated users and complex software environments, such as VS Code or Office, is currently too complicated. Directly feeding real environmental states to simulated users doesn't effectively aid in simulating user activities, which is our primary focus.
> > > Thus, we simulate environmental states during data generation (also reduce the complexity for models to simulate user activities) and utilize real environment states only in the test set, which is entirely collected from real user environments.
> > >
> > > We hope this clarifies our approach and addresses your concerns.
> > >
> > >
> > > # Q2: Observed High Recall
> > > As we calculate recall using the formula shown in the appendix:
> > > $Recall = \frac{TP}{TP+FN}$
> > >
> > > Here, TP (True Positives) is defined as instances where the model predicts tasks and the user accepts them.
> > > As discussed in Section 4.2, current models tend to provide excessive assistance, even when it is unnecessary. This results in a high recall, which contributes to a significant false-alarm issue.
> > > We experimented with various prompts before settling on the one shown in the appendix. It was surprising to discover how challenging it is to reduce a model's proactive assistance, as they still tend to offer more help than needed.
> > > Below is the number of tasks proposed by different models:
> > > | Model             | Task Proposed |
> > > |-------------------|---------------|
> > > | GPT-4o-mini       | 220           |
> > > | GPT-4o            | 212           |
> > > | Claude-3.5-Sonnet | 201           |
> > >
> > > These high numbers contribute to elevated TP values.
> > > FN (False Negatives) is defined as instances where the model does not predict tasks, but the user needs one. Due to the excessive assistance from current models, the condition where the model fails to predict tasks is rarely met.
> > > All these factors combined result in a high recall.
> > >
> > >
> > > We hope this clarification addresses your concerns and look forward to any further feedback or questions you might have. Your insights are invaluable in helping us refine our approach.

---

> ### Comment · Reviewer_fw7i · 2024-11-27
>
> Based on your explanation, it appears that the model's tendency to 'overly assist' is the primary factor leading to the gain, rather than the effect of the proposed framework? In other words, it is "good for misconceived reasons".

---

> > ### Author Response · Authors · 2024-11-27
> > **Clarification For Evaluation**
> >
> > Thank you for your feedback.
> >
> > # Q. Evaluation metrics
> >
> > Recall is a useful metric for identifying whether the model can detect the user's needs. However, as we previously described, Recall is not ideal for our task because models tend to propose excessive assistance. This behavior leads to a high False Alarm(the proportion of rejected tasks among all proposed tasks) ratio, while maintaining a lower FN rate. As a result, most models achieve a high Recall score.
> >
> > The False Alarm metric is used to better evaluate the extent of excessive assistance. As shown in the table below, the fine-tuned models exhibit a lower False Alarm rate, indicating that they are better at proposing assistance only when appropriate.
> >
> > To balance the trade-off between proposing too much and proposing too little, we selected the F1-Score as our final evaluation metric. The finetuned model shows both a higher Recall Score and a significant improvement in F1-Score, primarily driven by the reduction in False Alarm.
> >
> >
> > | Model                  | Recall  | Precision | Accuracy | False-Alarm | F1-Score  |
> > |:----------------------:|:-------:|:---------:|:--------:|:-------------:|:-----------:|
> > | LLaMA-3.1-8B           | 98.86%  | 38.16%    | 39.06%   | 61.84%      | 55.06%    |
> > | LLaMA-3.1-8B-Proactive | 99.06%  | 49.76%    | 52.86%   | **50.24%**      | **66.25%**    |
> > | Qwen2-7B               | 98.02%  | 44.00%    | 43.61%   | 56.00%      | 60.74%    |
> > | Qwen2-7B-Proactive     | 100.00% | 49.78%    | 50.66%   | **50.22%**      | **66.47%**    |

---

> ### Author Response · Authors · 2024-11-29
> **Kindly Request for You Feedback Before Rebuttal Deadline**
>
> We are ready to address any remaining misunderstandings or confusion you may have. We would like to know if our explanation helped resolve your concerns regarding our evaluation and the metrics used. If there are still any unresolved issues, we kindly ask for your feedback.
>
> As the rebuttal deadline is approaching, we would greatly appreciate any further suggestions, concerns, or comments at your earliest convenience. We are eager to continue the discussion and clarify any unsolved points to ensure all matters are fully addressed.
>
> Thank you again for your time and consideration.

---

> ### Author Response · Authors · 2024-12-02
> **Follow-Up on Rebuttal Feedback**
>
> We truly appreciate the time and effort you've dedicated to reviewing our paper and sharing your insights.
>
> As the rebuttal deadline approaches, we wanted to touch base and see if there are any lingering questions or concerns we can address. We’re eager to continue the conversation and would love to hear any additional thoughts or suggestions you might have.
>
> Thank you once again for your valuable feedback. We look forward to your response!

---

### Official Review · Reviewer_ZTiB · 2024-10-29

**Soundness:** 3
**Presentation:** 3
**Contribution:** 2
**Rating:** 6
**Confidence:** 4

**Summary:**

The paper provides a novel approach towards building proactive LLMs. To this end, the authors propose a data collection pipeline on three different types of tasks (coding, writing, daily life). The data pipeline uses Activity Watcher to collect human data (clicks, clipboard, etc.) on these different tasks.

GPT-4o is used to synthetically generate more events and create "ProactiveBench" dataset. They train a reward model to simulate human judgement from some human annotations and create an automatic evaluator of proactiveness. The results show that this reward model is better aligned in comparison to existing LLMs (LLaMa, GPT-4o).

They fine-tune two models (LLaMA, Qwen) on their Proactive Bench training set and show that this strategy leads to better proactiveness against baselines - measured in terms of precision, recall, accuracy, False-Alarm and F1-Score against the test set. Qwen2-7B based LLM fine-tuned on ProactiveBench achieves 66.47% F1 score, in comparison to 64.60% from GPT-4o.

They also perform ablation studies where they allow LLMs to output multiple candidates, and a dummy feedback  from their reward model to improve on its prediction - both strategies lead to some improvement in F1.

**Strengths:**

- The authors attempt to improve on a novel perspective and quality of LLM agents - proactiveness. The way they model this quality is also simple yet interesting - using rewards from user agents. They build a benchmark around this and show effectiveness of their approach towards building proactive agents.
- The design of the reward model and human judgement collection is thorough - they include all possible scenarios and the logic is sound.
- The additional metrics for the agreement ratio are interesting and help provide a better understanding of the performance of various reward models.
- The authors show improved performance against both open-source and closed-source models, when the LLMs are fine-tuned on their ProactiveBench dataset.
- The figures are really good, and they help with the understanding of the approach. The paper is more or less clear (see weaknesses - clarifications).

**Weaknesses:**

**General Weaknesses**
- GPT-4o already achieves 64.60% F1, while the author's best F1 score is 66.47%, which is a ~2% relative improvement. The test set (233 events) is not big enough for this difference to be statistically significant. Would be nice to have some discussion on why the authors feel that this improvement will have an impact on future research in this area, considering the fact LLMs keep getting better by the day.
- The experiment is performed against their reward-tuned model, and their own fine-tuned proactive agent, both on subsets of ProactiveBench (please correct me if I am wrong). This casts a doubt on whether actual users will benefit from the approach. Ideally, there should have been some user studies to support this, possibly with unseen events/scenarios.
- The authors do not discuss why they think Qwen performs better than LLaMa and GPT-4o.

**Limitations**
- The authors do not discuss the resource consumption for LLMs when they are proactively predicting tasks on existing systems. A good reason why most LLM agents today are reactive could be because of this issue - it is not possible to have an 7B parameter model constantly running on-device for proactive response generation.
- The authors discuss this in the limitations section - but the gym / agents are in a toy-setting with limited task types. The authors do not perform any real-user evaluations i.e. actually integrating the proactive agent into a system and getting an actual user to evaluate how helpful or disruptive the agent is.

**Missing clarifications/data**:
- Table 4: Can the proactive agents also be used with multiple candidates and with RM? If yes, then this table should contain results for those too.
-  The paragraph describing the human annotation in Section 3.5 could use improvement:
    - It is not clear what is the "cosine distance" computed against.
    - It is also not clear what "consistency rate" means or how is it measured.
    - What is the annotator agreement on train set?
- How is the GPT-4o explanation for the user judgement used downstream? Is this actually used?
- Section 3.4 - "it first updates its memory with the event" - what kind of memory mechanism is being used?
- Section 4.1 - Setting - How is the LLaMa 3.1 8B-Instruct trained? Are all the layers unfrozen during the training?
- Line 356 - "The temperature is set to 0 during testing." Which temperature? The one used for decoding? This line disrupts the flow of the reading, might be better placed elsewhere.
- Minor:
    - Not sure if I missed this: would be nice to have a well-defined description for what events and scenarios mean.
    - Would be nice to have examples of failure modes (showing scenarios, events, states, predictions), even if in Appendix, for different LLMs in comparison to authors' fine-tuned variant to clarify the differences.

**Questions:**

- They use GPT-4o for almost the entirety of the pipeline, except for initial user-data, and initial human judgement collection.
	- How do they ensure that the created scenarios/events and states make logical sense, if they are doing it at scale?

- Not relevant to the paper's strengths/weaknesses: What are the authors' thoughts on being able to tune reactiveness and proactivness? Each user and/or task may have different preferences, is there a straight forward way to control this?

- Line 408 - "In short, while most models can assist when needed, they still frequently offer unnecessary help, even when instructed to provide only essential assistance."
	- Do the authors have any thoughts on how this can be prevented in the future?

- How many events are collected in real-world events? How much is synthetically generated?

- How long are the context windows for the events i.e how many events are used for a single prediction?

- To clarify, the judgement is the only thing that is done from humans? And actions are produced by GPT-4o?

---

> ### Author Response · Authors · 2024-11-22
> **Clarification of evolving environmental states with GPT models; Tuning the Proactiveness; Reduce the False Alarm; Dataset Statics**
>
> # W3. Clarification of evolving environmental states with GPT models.
>
> The primary goal of integrating evolving environmental states with GPT models is to generate more coherent and logically flowing events, rather than achieving precise state change modeling. These states act as contextual reminders within our environmental simulations, aiding the model in crafting new events. The core purpose of these states is to introduce variability in the events generated by the model, enabling it to anticipate "what might happen next" or consider "what unusual events could occur."
>
> One of the main challenges in our approach is not the model's inability to accurately simulate an environment, but rather its tendency to create overly perfect or neutral scenarios. The real world is in chaos. Users often perform seemingly random actions, such as clicking aimlessly, moving the mouse erratically, or hesitating over words they can't spell, leading to a plethora of confusing actions. Current models struggle to replicate this unpredictability.
>
> By incorporating detailed states and aligning generated events with real-world occurrences, we've found that our simulation environment can introduce a degree of uncertainty and reality. For instance:
> ```text
> {"time": "10-18 09:17:40", "content": "User inputs 'ctrl_l' with mouse interactions, clicks in an application."}
> {"time": "09-17 15:46:53", "content": "The user double-checks the warranty information for the shortlisted devices before making a purchase."}
> {"time": "10-05 14:26:56", "content": "The user uses 'ctrl+c' to copy data about material sustainability from EcoChic\u2019s website for documentation."}
> ```
>
> The final performance gain also demonstrates the effectiveness of our method.
>
> Last But Not Related To The Question: from our experience of trying to obtain events that simulate the real world, we assume that the final false alarm could be further reduced by incorporating more real-world chaos events into our training data. We also believe how to obtain random and uncertainty from models is a critical question to study in the future.
>
> # Q2. How to tune proactiveness according to user preferences?
>
> In our current settings, we take the most conservative approach that only proposes tasks most people like to have, which might not be optimal in some settings.
> However, we can tune the agent's proactiveness by simply adding user feedback into historical events, like rejecting the proposal or instructions to the agent.
> Future studies should also consider these complex interactions between agents and users to automatically tune the proactiveness of the agent.
>
> # Q3. Ideas about how to reduce false alarms.
>
> This can be done in multiple ways:
> 1. Trying to collect more randomness from real users and improve the data generation pipeline could help. As we mentioned before,  the uncertainty and randomness of user activities or environment can not be perfectly modeled by current models, which causes a relatively high false alarm ratio for fine-tuned models.
> 2.  Simply collecting more real-world annotations for training agent models could also help.
> 3. Allowing the model to obtain more in-content samples of when they should or should not propose a query could also help.
>
>
> # Q4. Dataset Statics: how much real/synthetic data?
>
> Our dataset comprises four parts:
> 1. Dataset for the Agent Model
>   - Train set: 6,790 entries (Synthetic)
>   - Test set: 233 entries (Human-Collected)
> 2. Dataset for the Reward Model
>   - Train set: 1,640 entries (Human-Annotated)
>   - Test set: 120 entries (Human-Annotated)
>
> In total, there are around 700 real-world collected events and around 7,200 synthetic events.

---

> > ### Comment · Reviewer_ZTiB · 2024-11-27
> > **Response to Rebuttal - 1**
> >
> > > One of the main challenges in our approach is not the model's inability to accurately simulate an environment, but rather its tendency to create overly perfect or neutral scenarios. The real world is in chaos. Users often perform seemingly random actions, such as clicking aimlessly, moving the mouse erratically, or hesitating over words they can't spell, leading to a plethora of confusing actions. Current models struggle to replicate this unpredictability.
> >
> > This is an interesting insight! Thanks for providing clarification on this question.
> >
> > > Future studies should also consider these complex interactions between agents and users to automatically tune the proactiveness of the agent.
> >
> > Thanks for sharing this, I agree. It would be interesting to see how user feedback might be automatically used to tune the proactiveness.
> >
> > > This can be done in multiple ways
> >
> > Agree with all the pointers!
> >
> > > In total, there are around 700 real-world collected events and around 7,200 synthetic events.
> >
> > Thanks for clarifying this. I believe most of this was clear from a second reading of the paper, but would be nice to add a few words to enhance these differences if space permits.

---

> ### Author Response · Authors · 2024-11-22
> **Context Windows Problem; Proactive Interaction; Further Impact; User Studies**
>
> # Q5. Context windows for prediction.
>
> In our vanilla settings, we do not limit the number of events used for prediction.
> In our live demo, to improve responsiveness, we manually limit the number of events to 20.
>
> We also provide more performance comparisons for different settings here:
> | Model        | Settings | Recall  | Precision | Accuracy | False-Alarm |
> |--------------|----------|---------|-----------|----------|-------------|
> | LLaMA-3.1-8B | ws=1     | 100.00% | 35.53%    | 36.91%   | 64.47%      |
> |              | ws=3     | 98.73%  | 34.82%    | 36.91%   | 65.18%      |
> |              | ws=inf   | 100.00% | 36.52%    | 37.34%   | 63.48%      |
> | GPT-4o       | ws=1     | 94.94%  | 39.89%    | 49.79%   | 60.11%      |
> |              | ws=3     | 90.24%  | 40.66%    | 50.21%   | 59.34%      |
> |              | ws=inf   | 94.34%  | 49.51%    | 53.65%   | 50.50%      |
>
> The tables demonstrate that models might suffer a performance drop if not given sufficient contexts, for longer event context might let the model better capture the personality of the user.
>
>
> # Q6. Clarify proactive interactions.
> R. Yes, judgment is the only thing done by humans, and actions are conducted by agent models like GPT-4o.
> During our proactive interactions, the proactive agent proposes a task that the user might need. Then the user selects to accept or reject the proposal. If accepted, the proactive agent conducts actions to finish the proposed task. If rejected, the proactive agent does nothing.
> Hope this clarification can solve your question.
>
>
> # W1. Discuss the improvement made and further impact.
> The point of the comparison is to show that even a small model can proactively predict the user's intention after properly being finetuned rather than slightly outperforming GPT-4o. It suggests the effectiveness of our pipeline and training method.
>
> From that, the future impact should be proposing a way of constructing data and a new training method for the proactiveness of the agent, filling the lack of research field of Proactive Paradigm in this area for future research. The performance of the agent is not the main impact.
>
> # W2. User studies to support the claims of improved proactiveness.
>
> To clarify, our test set is collected independently from real-world users by tracking their actions from coding and writing scenarios, which is different from the training data generated from our environment gym. Our annotators annotate the events when they need help from the agent, and our reward model shows a significant agreement with our annotators. Due to that, what the gent proposed validated by the reward model should also be beneficial for the real user.
>
> Further user study is obviously a better way to demonstrate the superiority. But considering that user action is uncontrollable (and always varies between different users), the user study is very hard to conduct. We are indeed trying to build such a study, but it might not be available soon.

---

> > ### Comment · Reviewer_ZTiB · 2024-11-27
> > **Response to Rebuttal - 2**
> >
> > > The tables demonstrate that models might suffer a performance drop if not given sufficient contexts, for longer event context might let the model better capture the personality of the user.
> >
> > Thanks for sharing these results! I think these are very interesting - over longer contexts, I suspect these models might be able to fully capture the personality (high recall, low false alarm) of the user. Definitely looking forward to having these results in the paper/appendix.
> >
> > > Hope this clarification can solve your question.
> >
> > It does, thanks for this clarification.
> >
> > > user's intention after properly being finetuned rather than slightly outperforming GPT-4o
> >
> > I agree about the effectiveness of the pipeline. I am still skeptical as to why the authors think that over-time proactiveness might not be automatically present with higher performance (at least as far as evaluation on the benchmark is concerned) with large scale training. However, I agree the insights are interesting nonetheless, and I may understated the impact in my previous comment.
> >
> > > Due to that, what the gent proposed validated by the reward model should also be beneficial for the real user.
> > > We are indeed trying to build such a study, but it might not be available soon.
> > Thanks for sharing this. I agree that the reward model is a good proxy and it is hard to conduct controlled user studies for this case.
> > Looking forward to the next results if any, but I think this issue is addressed for me for the time being.

---

> ### Author Response · Authors · 2024-11-22
> **Performance Gap Clarification; Resource Consumption Discussion; User Evaluations;**
>
> # W3. Performance of Qwen2-7B and LLaMA-3.1-8B
>
> Thank you for your inquiry regarding the performance differences between Qwen2-7B and LLaMA-3.1-8B. We find our original downloaded LLaMA-3.1-8B model contains bugs that impact the final performance.
> We've updated the performance of LLaMA-3.1-8B to 55.06%, which is still a bit lower than Qwen2-7B.
>
> Upon further careful examination, we observed that Qwen2-7B provides more concrete and actionable responses, leading to higher accuracy and fewer false alarms. Our goal was to design tasks that are clear and user-friendly, and it appears that LLaMA-3.1-8B does not consistently meet this objective.
>
> Here are some comparisons to illustrate this point:
> ```text
> Event Timestamp: 1717335893.215
> Event Content: The user clicks on the web page, types 'swift ui list', and presses 'Enter'.
>
> Qwen2-7B proposal: Provide a brief explanation or guide on Swift UI List.
> LLaMA-3.1-8B Proposal: Explore possible tasks related to either Python or Swift UI list based on the user's interest.
>
>
> Analysis: LLaMA-3.1-8B's response is abstract and lacks actionable guidance, unlike Qwen2-7B, which offers a direct and relevant task.
>
> --------
> Event Timestamp: 1717338374.868
> Event Content: The user scrolls through the search results for 'sustainable fashion brands' on Google.
>
> Qwen2-7B proposal: Provide a brief summary of the top sustainable fashion brands from the search results.
> LLaMA-3.1-8B Proposal: Help the user create a knowledge graph using Python to visualize and connect their research on sustainable branding and fashion.
>
> Analysis: LLaMA-3.1-8B introduces unnecessary complexity with an abstract task, whereas Qwen2-7B remains focused and relevant.
>
> --------
> Event Timestamp:1717494492.445
> Event Content: The user scrolls and clicks on a research report page about biodegradable cups.
>
> Qwen2-7B proposal: Provide a summary of the key points from the research report about biodegradable cups, along with suggestions for further reading or related studies.
> LLaMA-3.1-8B Proposal: Create a new python project in Visual Studio Code and assist with setting up the environment.
>
> Analysis: LLaMA-3.1-8B misunderstands the user's action, resulting in an irrelevant proposal, while Qwen2-7B delivers a precise and helpful task.
> ```
>
> Hope this can solve your concerns.
>
> # L1. Discuss resource consumption for the proactive agent.
>
> We appreciate your insight on the resource consumption of the agent.
> Resource consumption is an important engineering question, although it is not within the scope of this paper.
> In this paper, we establish our experiments by running remote models, which can already achieve satisfying latency.
> Running locally or remotely does not impact the performance of proactiveness. As the services of LLMs get faster and cheaper, and more accessible to general humanity, we hold the belief that the transition from reactive and proactive will happen.
>
> Moreover, while large models like GPT or Claude perform well generally, there is a trend to train small models to beat the large models on specific downstream tasks. Models like Gemma-4B, Qwen-2B, MiniCPM3-4B consume less but could perform well when properly finetuned, and the capability of the small models only grows. With more techniques, resource consumption will no longer be a concern in a few years.
>
> # L2. Real user evaluations.
>
> The original human annotation is based on an events series, so from this point, the performance of the agent is already evaluated by a real human, it's just by replaying the event series rather than the ideal interactive evaluation process.
>
> The difference in the user action and familiarity with the project will influence the content and results of the agent(E.g. Some users may have a slow typing rate, and the agent will focus on proposing and implementing the content, interacting with the agent will further slower the typing of the users.). In order to control the variance, we have to use the static data collected to train the reward model, by replaying we could obtain the consistent rate of different annotators and, furthermore make the reward model simulate a real user better.
>
> Fortunately, we already developed more complicated software and demos suitable for different devices and platforms. These projects will help to enlarge our testing environment and lay a solid foundation for real, interactive, instant evaluation in the future. All these projects will be released soon.

---

> > ### Comment · Reviewer_ZTiB · 2024-11-27
> > **Response to Rebuttal - 3**
> >
> > > We've updated the performance of LLaMA-3.1-8B to 55.06%, which is still a bit lower than Qwen2-7B.
> >
> > Thanks for testing this out and updating the results before the review deadline.
> >
> > > Upon further careful examination, we observed that Qwen2-7B provides more concrete and actionable responses, leading to higher accuracy and fewer false alarms. Our goal was to design tasks that are clear and user-friendly, and it appears that LLaMA-3.1-8B does not consistently meet this objective.
> >
> > Thanks for sharing these insights. Please go ahead and add them in the paper/appendix. Looks like LLaMa is pretty bad with understanding intent and deciding on the next steps in comparison.
> >
> > > but could perform well when properly finetuned, and the capability of the small models only grows. With more techniques, resource consumption will no longer be a concern in a few years.
> >
> > All valid points. However, I think some evaluation on the amount of resources it might need at the moment could be helpful. I agree that this is beyond the scope of the paper since there are so many other things that the paper already discusses, but would be a useful insight.
> >
> > > Fortunately, we already developed more complicated software and demos suitable for different devices and platforms. These projects will help to enlarge our testing environment and lay a solid foundation for real, interactive, instant evaluation in the future. All these projects will be released soon.
> >
> > This is great! Thanks for sharing this.

---

> ### Author Response · Authors · 2024-11-22
> **Predict multiple candidates with RM; Human Annotation Details; GPT-4o Explanation Usage; Memory Mechanism; Training Settings; Temperature Settings Explanation**
>
> # M1. Predict multiple candidates with RM.
> Of course. Thanks for suggesting implementing our ablation study. The result is also updated in our paper. The section Ablation Study is renamed as Performance Analysis.
>
> | **Model**       | **Settings**      | **Recall** | **Precision** | **Accuracy** | **False-Alarm**  | **F1-Score** |
> |-----------------|-------------------|--------------------------|-----------------------------|-----------------------------|---------------------------------|-----------------------------|
> | GPT-4o-mini     | pred@1            | 100.00%                  | 35.28%                      | 36.12%                      | 64.73%                         | 52.15%                      |
> |                 | pred@3            | 99.32%                   | 65.32%                      | 66.52%                      | 34.68%                         | 78.80%                      |
> |                 | w/ RM             | 55.45%                   | 63.54%                      | 63.95%                      | 36.46%                         | 59.22%                      |
> |                 | pred@3, w/ RM     | 100.00%                  | 65.35%                      | 66.09%                      | 34.65%                         | 79.05%                      |
> | GPT-4o          | pred@1            | 98.11%                   | 48.15%                      | 49.78%                      | 51.85%                         | 64.60%                      |
> |                 | pred@3            | 100.00%                  | 63.56%                      | 64.81%                      | 36.44%                         | 77.72%                      |
> |                 | w/ RM             | 56.76%                   | 55.26%                      | 57.61%                      | 44.74%                         | 56.00%                      |
> |                 | pred@3, w/ RM     | 100.00%                  | 63.30%                      | 65.67%                      | 36.70%                         | 77.53%                      |
> | LLaMA-3.1-8B    | pred@1            | 98.86%                   | 38.16%                      | 39.06%                      | 61.84%                         | 55.06%                      |
> |                 | pred@3            | 100.00%                  | 52.79%                      | 52.79%                      | 47.21%                         | 69.10%                      |
> |                 | w/ RM             | 77.08%                   | 42.52%                      | 47.64%                      | 57.41%                         | 54.81%                      |
> |                 | pred@3, w/ RM     | 95.12%                   | 61.58%                      | 66.09%                      | 38.42%                         | 74.76%                      |
>
> In this table, predicting multiple candidates with rewarding (pred@3, w/ RM) achieves the best results for LLaMA-3.1-8B and GPT-4o-mini.
>
> # M2. Clarify the human annotation details.
> We obtain "cosine distance" by first embedding the predictions with `Text-Ada-Embedding` and calculating the cosine distance on the vectors.
> The consistency rate refers to the agreement ratio between human annotators.
> The agreement ratio on the train set for the reward model is around 77%.
> Thanks for your detailed feedback again! We've fixed it in our latest version.
>
> # M3. Usage of the GPT-4o explanation for the user judgment.
> The GPT-4o explanation is used as the CoT mechanism. During the training of the reward model, the explanation of the GPT-4o is used as a chain-of-thought for the judgment to help the reward model simulate a specific user better and perform better.
>
> # M4. Memory mechanism for the proactive agent.
> We adapt in-context short-term memory mechanisms that directly show the history since it is easy to implement and the memory history is linear, which aligns with the temporal event sequence.
>
> # M5. The training settings of the reward model.
> We conduct full-parameter finetuning. The detailed settings can be found in Section 4.1.
>
> # M6. Temperature Settings Explanation.
> Yes, the temperature is the decoding temperature. We emphasize the temperature here to show the result of our experiment is stable. We are sorry for any confusion, and we've updated the relative part to better clarify our ideas.

---

> > ### Comment · Reviewer_ZTiB · 2024-11-27
> > **Response to Rebuttal - 4**
> >
> > > In this table, predicting multiple candidates with rewarding (pred@3, w/ RM) achieves the best results for LLaMA-3.1-8B and GPT-4o-mini.
> >
> > Thanks for sharing these results and updating them in the paper.
> >
> > > The agreement ratio on the train set for the reward model is around 77%. Thanks for your detailed feedback again! We've fixed it in our latest version.
> >
> > I think this is a useful number that was missing, thanks for adding these.
> >
> > > We adapt in-context short-term memory mechanisms that directly show the history since it is easy to implement and the memory history is linear, which aligns with the temporal event sequence.
> >
> > Request you to please specify this on the paper/appendix.
> >
> > > We conduct full-parameter finetuning. The detailed settings can be found in Section 4.1.
> >
> > Apologies if I missed this earlier, thanks for clarifying.
> >
> > > Yes, the temperature is the decoding temperature. We emphasize the temperature here to show the result of our experiment is stable. We are sorry for any confusion, and we've updated the relative part to better clarify our ideas.
> >
> > Thanks for fixing this!

---

> ### Author Response · Authors · 2024-11-22
> **Events and Scenarios Definition; Failure Modes**
>
> # M7. Well-defined description of what events and scenarios mean.
> We're sorry for not making the definition clearer. To clarify, an event is a Time + Description of what happens. For example:
> ```text
> {
>     "time": "1717377997.0",
>     "event": "The user starts working on a file 'CheckName.rb' within a Ruby project in Visual Studio Code."
> }
> ```
> A scenario is a global review of what's going on. Specifically for our data generation settings, it contains environment descriptions, tasks that the user is performing, and user persona.
> We've updated related parts and provided more examples for clarity.
>
> # M8. Failure modes examples for comparison.
> Thank you for the valuable question. Here we give more failure modes analysis (we've also updated the appendix to help a better understanding.):
>
> ***Not A Task***: The model does not give a real task at all.
> ```text
> Observation: {
> "time": "1717335907.603",
> "event": "The user switches to Visual Studio Code."
> }
> GPT-4o-mini Response: It seems you may be switching to Visual Studio Code for development. If you need any coding tips, resources, or help with Swift UI in VS Code, feel free to ask!
> ```
> Under this event, the agent did not take the event as an action, but rather a call for chat. This chat is not a task we want.
>
>
> ***Unspecific Task***: The proposed task is non-executable due to a lack of a clear and detailed description.
> ```text
> Observation: {
> "time": "1717329066.416",
> "event": "The user performed a search for ’js fade in/out’ on
> Bing."
> }
> GPT-4o-mini Response: Compile a list of JavaScript resources, tutorials, and documentation that cover advanced topics related to the user’s search.
> ```
> Under this event, the model proposal is not clear (not pointing out the user's research content) and impractical (Compiling a list of resources, tutorials, and documentation, which is too much for the user and also out of the capability of the model).
>
>
> ***Irrelevant Task***: The proposed task is not related to the current context or meets the user's needs.
> ```text
> Observation: {
> "time": "1717336090.982",
> "event": "The user clicks, types ’list.swift’ in Visual Studio
> Code."
> }
> GPT-4o Response: I can help by providing tips on how to efficiently navigate and manage files in Visual Studio Code or offer assistance with the content of ’list.swift’.
> ```
> Under this event, the user created a new file in vscode. However, the agent provided a task about file management, which is irrelevant to the user's intention.
>
> ***Unwanted Proposal***: The user does not want to be interrupted.
> ```text
> Observation: {
> "time": "1717329211.15",
> "event": "The user opened ’http://127.0.0.1:5500/main.html’ in the web browser."
> }
> GPT-4o Response: Offer to help troubleshoot or verify the ’fade in fade out’ effect if it doesn’t work as expected.
> ```
> Under this context, since whether the edition is successful or not is not clear, the agent should remain silent rather than propose troubleshooting currently.
>
>
> If you have any additional questions or require further clarification, please feel free to reach out.

---

> > ### Comment · Reviewer_ZTiB · 2024-11-27
> > **Response to Rebuttal - 5**
> >
> > > We've updated related parts and provided more examples for clarity.
> > Thanks for fixing this. By the end of the paper I had a general idea of what these terms mean, but useful to have a clearer definition wherever possible.
> >
> > > Failure modes examples for comparison.
> >
> > Thanks for sharing these! I love these failure modes, would love to see them on the paper.

---

> ### Author Response · Authors · 2024-11-27
> **Request for Further Discussion Before Rebuttal Deadline**
>
> We sincerely appreciate your thoughtful feedback and suggestions on our work. We are eager to demonstrate that our work holds promise and greater value. To address the concerns you raised, we have provided clarifications on previously missing details, including performance analysis, datasets, memory mechanisms, training processes, and omitted concepts. Additionally, we have conducted new experiments on context windows and multiple tasks with the reward model, and we have shared our ideas on potential applications, resource consumption, and future improvements. The paper has been updated to incorporate these clarifications.
>
> As the rebuttal deadline approaches, we would like to know if our responses have sufficiently addressed your concerns and questions. We welcome any further feedback or discussion to help us continue improving our work. Thank you for your time and consideration.

---

> ### Comment · Reviewer_ZTiB · 2024-11-27
> **Response to Authors Comment**
>
> Apologies for the delay in responding to the rebuttal.
>
> I think the authors have addressed most of my concerns, and provided clarifications wherever I had doubts.
>
> I will retain my rating for the paper as I wanted to give them a 7, but I can only choose between 6 and 8.
>
> Overall, I think this is an interesting paper and would love to see it pushed through.

---

> > ### Author Response · Authors · 2024-11-28
> > **Response to Reviewer**
> >
> > We are more than happy to address all your concerns and issues, and we sincerely appreciate the time and effort you've dedicated to our paper. Your support and encouragement are invaluable to us. We will continue to improve our work in the future to better meet expectations.

---

> > ### Author Response · Authors · 2024-11-28
> > **Response to Reviewer**
> >
> > We sincerely appreciate your thoughtful review and understand the inconvenience in adjusting your rating. If possible, we would be grateful if you could consider increasing your confidence level for this paper. Your support would mean a great deal to us.

---

### Official Review · Reviewer_tUPL · 2024-11-04

**Soundness:** 2
**Presentation:** 2
**Contribution:** 2
**Rating:** 5
**Confidence:** 4

**Summary:**

This paper presents a proactive agent framework designed to predict human intentions based on real-world-inspired environmental observations. To support this, the authors introduce ProactiveBench, a benchmark dataset containing 6,790 events across coding, writing, and daily life activities, capturing user behaviors such as keyboard and mouse inputs, clipboard content, and browser activity.

The data generation process integrates three components: a proactive agent, a user agent, and an environment simulation to emulate realistic interactions. Leveraging human-annotated data, the authors train a reward model that simulates human judgment, achieving 91.8% consistency with human evaluations.

To further enhance the capability to proactively predict human intentions, the authors fine-tuned the LLaMA-3.1-8B and Qwen2-7B. Among these, Qwen2-7B-Proactive achieved the highest accuracy and F1-scores in anticipating user intentions, outperforming both open-source and closed-source baselines.

**Strengths:**

* The paper presents a well-structured framework for proactive agents, supported by robust experimental design.
* The creation of ProactiveBench provides a comprehensive benchmark dataset that will be highly valuable for future research in proactive language models and user-agent interaction studies.
* The paper employs reinforcement learning to simulate the user agent and model user preferences, enabling a realistic and dynamic approach to assess agent behaviors and refine their decision-making processes.

**Weaknesses:**

* This paper lacks sufficient detail in sections such as *Section 3.3: Environment Gym* and *Section 4.1: Reward Model Assessment*, which hinders a full understanding of the framework and evaluation process.
* The paper does not adequately discuss current research on real-world proactive agents or clarify how this work differs from existing approaches.
* The experimental analysis is limited, and the ablation study section is somewhat misleading.

**Questions:**

* It's unclear whether the environment gym relies on an interactive simulator or is simply text-based.
* The four stages used to describe the environment gym setup seem redundant and could be streamlined for clarity.
* More information is needed on the dataset used for reward model assessment, particularly whether the test data is out-of-domain or in-domain.
* The paper lacks information on the human annotators involved in determining the agreement ratio in the reward model assessment. It is better to provide details on their demographics or annotation guidelines.
* In Proactive Agent Evaluation, it is unclear why Qwen2-7B outperforms LLaMA-3.1-8B despite having fewer parameters. Additionally, the performance gains from fine-tuning on Qwen-7B appear relatively small, which should be further discussed.
* The term *ablation study* seems somewhat misleading here, as this section primarily compares public models in different settings rather than systematically removing or modifying components. Furthermore, it is unclear why the *Predict Multiple Tasks* setting improves overall performance, and additional explanation would be helpful.

---

> ### Author Response · Authors · 2024-11-22
> **Simulation Clarification; Stage Clarification; Reward Model Dataset; Annotation Details.**
>
> # W1 Q1. Interactive simulator or text-based simulator?
> To clarify, we build an environment gym to obtain training data for our agent model only. It's an event-driven asynchronous text-based simulator, like Text-World[1] or ALF-World[2]. The user agent and proactive agent in the gym are connected by the event transmitting mechanism in the gym.
>
> During running, the gym will dynamically update the environmental states based on the events generated by the user agent. The proactive agent monitors the behavior and states of the environment to produce proactive predictions.
>
>
> 1. Côté, Marc-Alexandre, et al. "Textworld: A learning environment for text-based games." Computer Games: 7th Workshop, CGW 2018, Held in Conjunction with the 27th International Conference on Artificial Intelligence, IJCAI 2018, Stockholm, Sweden, July 13, 2018, Revised Selected Papers 7. Springer International Publishing, 2019.
> 2. Shridhar, Mohit, et al. "Alfworld: Aligning text and embodied environments for interactive learning." arXiv preprint arXiv:2010.03768 (2020).
>
>
> # W1 Q2. The four stages of the environment gym setup seem redundant and could be streamlined for clarity.
>
> The four stages represent different perspectives of our data generation.
> To calibrate the event generated by the gym, we first collect real-world events in the user's environments to serve as seed events. Then to improve the consistency and detail of the simulation, we introduce the scenario generation stage to create scenarios with different environmental objects, like the operating system version, applications installed, the running apps, etc. After that, we iteratively generate new events from the gym and update the environment states in the event generation stage and state maintenance stage respectively.
>
> Hope this clarification can solve your question. A more detailed explanation of the possible redundant would be appreciated to better locate the problem.
>
>
> # W1 Q3. More information about the dataset used for reward model assessment.
> To collect the data, we develop annotation tools (VSCode add-ons, browser extensions, general user input monitoring, system monitoring, interactive panels) to collect real user activities and generate proactive prediction with 9 models (Cluade-3-Sonnet-20240229, Claude-3.5-Sonnet-20240620, GPT-3.5-Turbo, GPT-4, GPT-4o-mini, GPT-4o, LLaMA-3-8B-Instruct, LLaMA-3-70B-Instruct, Mixtral-8x7B). We manually annotated each prediction given at least 3 times to get a major voted judgment, resulting in a total of 1,760 data entries (as we described in Section 4.1). That is, the entire dataset to train and evaluate the reward model is human-annotated.
>
> We randomly sample the split the dataset into train (1,640 entries) and test (120 entries) set of the reward model. The evaluation of the reward model is conducted in-domain.
> Besides, we've also updated Table 1 to help better understand our dataset.
>
>
> # W1 Q4. Annotator demographics and Annotation Guidelines.
>
> The annotators' average age is 22.7 (12 people in total). Most of them study computer science for a bachelor's degree. All annotators read in English.
> Here are the Annotation Guidelines we used during annotation:
>
> ```text
> In this annotation task, you need to observe two elements:
> The sequence of events is described in the Observations module, where each event has an event field detailing what occurred.
>
> Tasks generated by the model in the Tasks module are related to the current event sequence.
>
> You need to carefully review the five tasks provided by the model. Analyze which tasks align with your potential needs in the context of the described event sequence, without excessively disrupting your normal workflow. Then, select the corresponding tasks in the checkboxes below. For example, if you believe that in the above example, tasks 3 and 4 are what you expect the agent to perform, and the others are not, then check the boxes for tasks 3 and 4.
>
> If you think none of the tasks are appropriate (or you do not want to be disturbed by the agent in the current situation), please select the reject all checkbox.
>
> After making your selection, click the continue button to submit the annotation results. At this point, new events will be updated in the observations, and you will repeat the above process until the entire event sequence is annotated.
>
> ### Special Events
> In addition to regular task annotations, some scenarios offer two choices, requiring you to determine whether help is needed given the observation. In this case, you need to choose between task1 and task2 (note: do not select reject all).
> ```

---

> ### Author Response · Authors · 2024-11-22
> **Real-world Proactive Agents; Performance Gap Clarification; Additional explanation on Predict Multiple Tasks.**
>
> # W2. Discussion on real-world proactive agents.
>
> We have identified a lack of discussion about real-world proactive agents. We've updated the paper's related work section to better clarify the differences.
> Please refer to the latest version for details.
>
> # Q5. Performance gaps between Qwen2-7B and LLaMA-3.1-8B
>
> Thank you for your inquiry regarding the performance differences between Qwen2-7B and LLaMA-3.1-8B. We find our original downloaded LLaMA-3.1-8B model contains bugs that impact the final performance.
> We've updated the performance of LLaMA-3.1-8B to 55.06%, which is still a bit lower than Qwen2-7B.
> Upon further careful examination, we observed that Qwen2-7B provides more concrete and actionable responses, leading to higher accuracy and fewer false alarms. Our goal was to design tasks that are clear and user-friendly, and it appears that LLaMA-3.1-8B does not consistently meet this objective.
>
> Here are some comparisons to illustrate this point:
> ```text
> Event Timestamp: 1717335893.215
> Event Content: The user clicks on the web page, types 'swift ui list', and presses 'Enter'.
>
> Qwen2-7B proposal: Provide a brief explanation or guide on Swift UI List.
> LLaMA-3.1-8B Proposal: Explore possible tasks related to either Python or Swift UI list based on the user's interest.
>
>
> Analysis: LLaMA-3.1-8B's response is abstract and lacks actionable guidance, unlike Qwen2-7B, which offers a direct and relevant task.
>
> --------
> Event Timestamp: 1717338374.868
> Event Content: The user scrolls through the search results for 'sustainable fashion brands' on Google.
>
> Qwen2-7B proposal: Provide a brief summary of the top sustainable fashion brands from the search results.
> LLaMA-3.1-8B Proposal: Help the user create a knowledge graph using Python to visualize and connect their research on sustainable branding and fashion.
>
> Analysis: LLaMA-3.1-8B introduces unnecessary complexity with an abstract task, whereas Qwen2-7B remains focused and relevant.
>
> --------
> Event Timestamp:1717494492.445
> Event Content: The user scrolls and clicks on a research report page about biodegradable cups.
>
> Qwen2-7B proposal: Provide a summary of the key points from the research report about biodegradable cups, along with suggestions for further reading or related studies.
> LLaMA-3.1-8B Proposal: Create a new python project in Visual Studio Code and assist with setting up the environment.
>
> Analysis: LLaMA-3.1-8B misunderstands the user's action, resulting in an irrelevant proposal, while Qwen2-7B delivers a precise and helpful task.
> ```
>
> Here we also provide the updated performance table:
>
> | Model                  | Recall  | Precision | Accuracy | False-Alarm | F1-Score  |
> |------------------------|---------|-----------|----------|-------------|-----------|
> | GPT-4o-mini            | 100.00% | 35.28%    | 36.12%   | 64.73%      | 52.15%    |
> | GPT-4o                 | 98.11%  | 48.15%    | 49.78%   | 51.85%      | 64.60%    |
> | Claude-3.5-Sonnet      | 97.89%  | 45.37%    | 49.78%   | 54.63%      | 62.00%    |
> | LLaMA-3.1-8B           | 98.86%  | 38.16%    | 39.06%   | 61.84%      | 55.06%    |
> | LLaMA-3.1-8B-Proactive | 99.06%  | 49.76%    | 52.86%   | 50.24%      | 66.25%    |
> | Qwen2-7B               | 98.02%  | 44.00%    | 43.61%   | 56.00%      | 60.74%    |
> | Qwen2-7B-Proactive     | 100.00% | 49.78%    | 50.66%   | 50.22%      | 66.47%    |
>
> Hope this can solve your concerns :).
>
> # W3 Q6. Ablation study and additional explanation on Predict Multiple Tasks.
>
> Thank you for the suggestion. We've renamed this section to "Performance Analysis" to avoid any misunderstanding.
> We also update Table 4 in the paper to more clearly present the results.
> As we described in Section 4.3, the Predict Multiple Tasks (pred@3) settings allow the model to generate multiple candidate proactive tasks. This means the model can generate more diverse proactive tasks and have more chances of being accepted (though it might increase the user's burden to read tasks). As a result, the model's performance improved with more acceptance.

---

> > ### Author Response · Authors · 2024-11-22
> > **Updated Table 4 for More Comparison.**
> >
> > We also update Table 4 in the re-versioned paper to avoid misunderstanding.
> >
> > | **Model**       | **Settings**      | **Recall**| **Precision** | **Accuracy** | **False-Alarm**  | **F1-Score**|
> > |-----------------|-------------------|--------------------------|-----------------------------|-----------------------------|---------------------------------|-----------------------------|
> > | GPT-4o-mini     | pred@1            | 100.00%                  | 35.28%                      | 36.12%                      | 64.73%                         | 52.15%                      |
> > |                 | pred@3            | 99.32%                   | 65.32%                      | 66.52%                      | 34.68%                         | 78.80%                      |
> > |                 | w/ RM             | 55.45%                   | 63.54%                      | 63.95%                      | 36.46%                         | 59.22%                      |
> > |                 | pred@3, w/ RM     | 100.00%                  | 65.35%                      | 66.09%                      | 34.65%                         | 79.05%                      |
> > | GPT-4o          | pred@1            | 98.11%                   | 48.15%                      | 49.78%                      | 51.85%                         | 64.60%                      |
> > |                 | pred@3            | 100.00%                  | 63.56%                      | 64.81%                      | 36.44%                         | 77.72%                      |
> > |                 | w/ RM             | 56.76%                   | 55.26%                      | 57.61%                      | 44.74%                         | 56.00%                      |
> > |                 | pred@3, w/ RM     | 100.00%                  | 63.30%                      | 65.67%                      | 36.70%                         | 77.53%                      |
> > | LLaMA-3.1-8B    | pred@1            | 98.86%                   | 38.16%                      | 39.06%                      | 61.84%                         | 55.06%                      |
> > |                 | pred@3            | 100.00%                  | 52.79%                      | 52.79%                      | 47.21%                         | 69.10%                      |
> > |                 | w/ RM             | 77.08%                   | 42.52%                      | 47.64%                      | 57.41%                         | 54.81%                      |
> > |                 | pred@3, w/ RM     | 95.12%                   | 61.58%                      | 66.09%                      | 38.42%                         | 74.76%                      |
> >
> > We have updated Section 4.3 for better clarity.
> > If you have any additional questions or require further clarification, please feel free to reach out.

---

> ### Author Response · Authors · 2024-11-27
> **Request for Further Discussion Before Rebuttal Deadline**
>
> We greatly appreciate your thoughtful feedback and have made every effort to address your concerns and questions. So far, we have clarified details regarding our simulator, four-stage generation process, annotator demographics and guidelines, reward model datasets, performance analysis (formerly the Ablation Study section), and the performance gap between Qwen2-7B and LLaMA-3.1-8B. Additionally, we have updated our paper to highlight the absence of real-world proactive agents and to provide a more comprehensive performance analysis.
>
> As the rebuttal deadline is approaching, we would greatly value your further thoughts and feedback to help us improve and better explain our work. Your patience and understanding mean a great deal to us, and we would be grateful if you could share any additional questions or concerns you may have. Thank you once again for your time and consideration.

---

> > ### Comment · Reviewer_tUPL · 2024-11-28
> > **Official Comment by Reviewer tUPL**
> >
> > Thank you for the clarification. Regarding your response to Question 5, it seems unusual that every proposal from LLaMA-3.1-8B involves the use of Python to complete the task, regardless of context. Could you provide further explanation on this?

---

> > > ### Author Response · Authors · 2024-11-28
> > > **Clarification on the Anomalies Observed in LLaMA-3.1-8B**
> > >
> > > You raised an important concern regarding the fact that all selected cases from LLaMA-3.1-8B involve the use of Python to complete tasks. This anomaly arises due to the model’s tendency to overemphasize remote or irrelevant context, regardless of the actual task at hand. One key distinction between generated data and real-world data is the presence of noise in real-world data, which requires the model to effectively filter out irrelevant information and focus on the user’s actual intent. LLaMA-3.1-8B struggled with this differentiation, which led it to generate Python-related tasks.
> > >
> > > Upon investigation, we discovered that some information about Python was recorded in the event history. LLaMA-3.1-8B misinterpreted this information, assuming that the user intended to download or work with Python, and consequently generated task suggestions related to it. For the tasks at hand, LLaMA-3.1-8B incorrectly associated them with Python, further compounding its prediction errors. To provide additional clarity, we have included the reasoning process of the LLaMA-3.1-8B model below.
> > >
> > > ```text
> > > Event Timestamp: 1717335893.215
> > > Event Content: The user clicks on the web page, types 'swift ui list', and presses 'Enter'.
> > > LLaMA-3.1-8B's Reasoning: Based on the user's previous action, they were trying to install Python, but now they are interested in Swift UI list. This seems like a change in topic or goal. I should consider providing a task related to either Python or Swift UI list, depending on the user's current needs.
> > > LLaMA-3.1-8B's Proposal: Explore possible tasks related to either Python or Swift UI list based on the user's interest.
> > >
> > > Event Timestamp: 1717338374.868
> > > Event Content: The user scrolls through the search results for 'sustainable fashion brands' on Google.
> > > LLaMA-3.1-8B's Reasoning: Based on the previous events, the user has been researching sustainable branding and sustainable fashion. They seem to be doing research and gathering information. I can provide a task that helps the user organize their findings or find more relevant information.
> > > LLaMA-3.1-8B's Proposal: Since you're researching sustainable fashion brands, I can help you create a knowledge graph using Python to visualize and connect your research on sustainable branding and fashion. Would you like me to guide you through this process?
> > >
> > > Event Timestamp: 1717494492.445
> > > Event Content: The user scrolls and clicks on a research report page about biodegradable cups.
> > > LLaMA-3.1-8B's Reasoning: The user has been struggling with setting up the python environment and searching for related topics. They seem to be taking a break and focusing on reading a report about biodegradable cups. However, considering their previous difficulties and the lack of clear resolution, I think they may still need help with their original task.
> > > LLaMA-3.1-8B's Proposal: Create a new python project in Visual Studio Code and assist with setting up the environment.
> > > ```
> > >
> > > This issue highlights one of the reasons why Qwen2-7B outperforms LLaMA-3.1-8B. Qwen2-7B demonstrates a stronger ability to focus on the current task and shows greater resilience to noisy or irrelevant information, whereas LLaMA-3.1-8B is more easily influenced by such distractions.

---

> > > ### Author Response · Authors · 2024-11-28
> > > **Further explanation on failure modes**
> > >
> > > Below we also provide some common failure modes:
> > >
> > > **Not A Task**: The model does not give a real task at all.
> > > ```json
> > > "observation": {
> > >     "time": "1717335907.603",
> > >     "event": "The user switches to Visual Studio Code."
> > > },
> > > "gpt-4o-mini": [
> > >     "It seems you may be switching to Visual Studio Code for development. If you need any coding tips, resources, or help with Swift UI in VS Code, feel free to ask!"
> > > ]
> > > ```
> > > Under this event, the agent did not take the event as an action, but rather a call for chat. This chat is not a task we want.
> > >
> > > **Unspecific Task**: The proposed task is non-executable due to a lack of a clear and detailed description.
> > > ```json
> > > "observation": {
> > >     "time": "1717329066.416",
> > >     "event": "The user performed a search for 'js 淡入淡出' on Bing."
> > > },
> > > "gpt-4o-mini": [
> > >     "Compile a list of JavaScript resources, tutorials, and documentation that cover advanced topics related to the user's search."
> > > ]
> > > ```
> > > Under this event, the model proposal is not clear (not pointing out the user's research content) and impractical (Complile a list of resources, tutorials, documentations, which is too much for the user and also out of the capability of the model).
> > >
> > > **Irrelevant Task**: The proposed task is not related to the current context or meets the user's needs.
> > > ```json
> > > "observation": {
> > >     "time": "1717336090.982",
> > >     "event": "The user clicks, types 'list.swift' in Visual Studio Code."
> > > },
> > > "gpt-4o": [
> > >     "I can help by providing tips on how to efficiently navigate and manage files in Visual Studio Code or offer assistance with the content of 'list.swift'."
> > > ]
> > > ```
> > > Under this event, the user created a new file in vscode. However, the agent provided a task about file management, which is irrelevant to the user's intention.
> > >
> > > **Unwanted Proposal**: The user does not want to be interrupted.
> > > ```json
> > > "observation": {
> > >     "time": "1717329066.416",
> > >     "event": "The user performed a search for 'js fade in fade out' on Bing."
> > > },
> > > //[hidden]
> > > "observation": {
> > >     "time": "1717329172.182",
> > >     "event": "The user switched to editing 'animation.js' in the project '/c:/Users/admin/code/js-animation'."
> > > },
> > > //[hidden]
> > > "observation": {
> > >     "time": "1717329211.15",
> > >     "event": "The user opened 'http://127.0.0.1:5500/main.html' in the web browser."
> > > },
> > > "gpt-4o": "Offer to help troubleshoot or verify the 'fade in fade out' effect if it doesn't work as expected."
> > > ```
> > >
> > > Under this context, since whether the edition is successful or not is not clear, the agent should remain silent rather than propose troubleshooting currently.
> > >
> > > If you have any additional questions about failure proposals or need further clarification, please feel free to reach out.

---

> ### Author Response · Authors · 2024-11-29
> **Kindly Request for You Feedback Before Rebuttal Deadline**
>
> We are fully prepared and eager to engage in a more in-depth discussion with you. We've provided clarification regarding the anomaly of LLaMA-3.1-8B and presented additional failure modes to further demonstrate the performance of different models. We would like to know if our explanation addressed your concerns.
>
> If you still have any remaining questions or uncertainties, we would be more than happy to assist in addressing them as soon as possible. We look forward to hearing from you and continuing the discussion!

---

> ### Author Response · Authors · 2024-12-02
> **Follow-Up on Rebuttal Feedback**
>
> Thank you for your thoughtful review of our paper. We truly value the insights you have provided. As the deadline for the rebuttal approaches, we wanted to reach out and check if there are any additional questions or concerns we can address.
>
> We are keen to continue the discussion and would greatly appreciate any further feedback or suggestions you may have. Please feel free to share your thoughts, and we will respond promptly.
>
> Thank you once again for your time and consideration.

---

### Official Review · Reviewer_9sAE · 2024-11-09

**Soundness:** 3
**Presentation:** 3
**Contribution:** 3
**Rating:** 6
**Confidence:** 3

**Summary:**

This paper addresses the limitations of reactive agent systems by introducing a proactive approach to LLM agents. The authors develop a data-driven method where real-world human activities are collected to generate task predictions, which are then labeled as accepted or rejected by human annotators. These labels train a reward model that simulates human judgment, enabling the agent to anticipate and initiate actions without explicit instructions. The authors create ProactiveBench, a dataset of 6,790 events, and show that fine-tuning agents with this dataset significantly enhances their proactive capabilities. It marks a promising step forward in human-agent collaboration.

**Strengths:**

I really appreciate the concept of proactive agents, as it addresses a crucial challenge in the planning field: actions should have long-term impacts—a factor often overlooked in prior research. This approach could inspire future agent research to prioritize anticipating future outcomes before choosing actions in the current step. I'm also impressed by the model's performance on ProActiveBench, where it significantly outperforms the GPT series models.

**Weaknesses:**

The main weaknesses lie in several points:

First, the training process for the proactive agent is not fully explained, particularly regarding the type of annotations used. Besides, while the introduction mentions human-annotated data, it may also cover some portions potentially synthesized automatically. It’s unclear what instructions human annotators followed, how annotation quality was ensured, and how quality control was managed for automatically generated data.

Additionally, the process of maintaining environment state requires further detail. Since this involves using GPT models to generate new states, it raises questions about the accuracy and reliability of GPT in simulating these states. More clarification here would be valuable.

I also have concerns about the comprehensiveness of the tested baselines. For instance, the GPT-based baselines might not fully leverage GPT's potential. Incorporating more chain-of-thought reasoning could lead to more grounded and reliable action predictions, similar to a world model that simulates multiple steps ahead. It would also be insightful to see a comparison showing performance without any proactive agents as a benchmark.

Additionally, the evaluation relies heavily on an in-house test set, raising questions about the broader applicability of the proactive agent approach. A more comprehensive evaluation, possibly across varied scenarios, would help clarify its generalizability and robustness.

**Questions:**

See above. I may also suggest incorporating more examples in the method part.

---

> ### Author Response · Authors · 2024-11-22
> **Response to W1, detailed data examples and quality control process**
>
> Thank you for your detailed review of our work. We really appreciate your approval of the concept of proactive agents, as it checks what could been done before conducting real actions.
> Here is our response to your concern:
>
> W1. The training process of the proactive agent is not fully explained & How quality control was managed for automatically generated data.
>
> In short, we obtain a reward model by collecting real world events and generating predictions with 9 different models (Cluade-3-Sonnet-20240229, Claude-3.5-Sonnet-20240620, GPT-3.5-Turbo, GPT-4, GPT-4o-mini, GPT-4o, LLaMA-3-8B-Instruct, LLaMA-3-70B-Instruct, Mixtral-8x7B). We manually annotated each prediction at least 3 times to get major voted judgement, resulting in a total of 1,760 data entries (as we deciped in Section 4.1). That is, the entire dataset to train and evaluate the reward model is human annotated.
> After that, we obtain an agent model by automatically generate traininng data with the reward model and use the generated data to finetune the models like LLaMA-3.1-8B-Instruct and Qwen2-7B.
> Here is one example of the data generated:
> ```text
> System
> ______
> <Role> You are a helpful assistant that provides proactive suggestions to the user. </Role>
>
> <Task> Understand what the user is doing and anticipate their needs based on events. Only propose assistance when you fully understand the user's actions. Use available operations to ensure the task is feasible. Execute the task if the user accepts your proposal. </Task>
>
> <Format> Respond in the following JSON format:
> {
>     "Purpose": "The purpose of the user's last action.",
>     "Thoughts": "Your thoughts on the user's actions.",
>     "Proactive Task": "Describe your proposed task, or set to `null` if no assistance is needed.",
>     "Response": "Inform the user about your assistance if proposing a task."
> }
> </Format>
>
> <Rules>
> - Ensure the proposed task is relevant to the events. - Focus on the user's current needs and predict helpful tasks.
> - Consider the timing of events.
> - Only offer proactive assistance when necessary.
> - Deduce the user's purpose and whether they need help based on event history.
> - Set `Proactive Task` to `null` if the user doesn't need help.
> </Rules>
>
> User
> ----
> {
>     "Instructions": "Now analyze the history events and provide a task if you think the user needs your help.",
>     "Observations": [
>        //....
>        {
>             "Time": "10-12 20:00:43",
>             "Event": "The user added edge cases to the Postman test suite to ensure robustness."
>         },
>         {
>             "Time": "10-12 20:01:18",
>             "Event": "The user monitored the output logs from the API to trace any potential errors during tests."
>         },
>         {
>             "Time": "10-12 20:01:48",
>             "Event": "The user noted the results of the tests, including any failed cases, and prepared to rectify them."
>         }
>     ]
> }
>
> Agent
> -----
> {
>     "Purpose": "The user is actively conducting thorough manual testing on their Flask API, focusing on executing tests and documenting the results of those tests.",
>     "Thoughts": "The user appears to be confident in their testing process and is already engaged in troubleshooting the noted errors. They have been utilizing multiple resources to resolve issues, indicating a proactive and self-sufficient approach to their development tasks. Since they feel in control and are making progress, they may not need further assistance right now.",
>     "Proactive Task": null,
>     "Response": ""
> }
> ```
>
> During the construction of the agent proposal, we randomly sampled the response from the model. To control the quality of the generated response, we evaluate the generated responses with the reward model and filter the responses rejected by the reward model (with over 70% of responses filtered), only saving the valid data as training data.
>
> We conduct full parameters supervised fine-tuning for the model we selected. We employ a total batch size of 32, a learning rate of 1e − 5, and an Adam Optimizer with a 0.1 warm-up ratio. We train the model for 3 epochs to prevent it from over-fitting. We use 8 A100-40G GPUs on one node to train for approximately 2 hours.

---

> ### Author Response · Authors · 2024-11-22
> **Annotation Guidelines, Clarification of evolving environmental states with GPT models.**
>
> # W2. Annotation Guidelines and Quality Control
>
> Here are the annotation guidelines (we've also updated it in the appendix):
> ```text
> In this annotation task, you need to observe two elements:
> The sequence of events is described in the Observations module, where each event has an event field detailing what occurred.
>
> Tasks generated by the model in the Tasks module are related to the current event sequence.
>
> You need to carefully review the five tasks provided by the model. Analyze which tasks align with your potential needs in the context of the described event sequence, without excessively disrupting your normal workflow. Then, select the corresponding tasks in the checkboxes below. For example, if you believe that in the above example, tasks 3 and 4 are what you expect the agent to perform, and the others are not, then check the boxes for tasks 3 and 4.
>
> If you think none of the tasks are appropriate (or you do not want to be disturbed by the agent in the current situation), please select the reject all checkbox.
>
> After making your selection, click the continue button to submit the annotation results. At this point, new events will be updated in the observations, and you will repeat the above process until the entire event sequence is annotated.
>
> ### Special Events
> In addition to regular task annotations, some scenarios offer two choices, requiring you to determine whether help is needed given the observation. In this case, you need to choose between task1 and task2 (note: do not select reject all).
> ```
>
> Each entry was annotated at least 3 times before we made it part of the final dataset. We check the agreement ratio of each annotator with the rest, filtering the data with a low agreement ratio. The final agreement ratio for the reward model test set is over 90%, demonstrating the reliability of our annotations.
>
>
> # W3. Clarification of evolving environmental states with GPT models.
>
> The primary goal of integrating evolving environmental states with GPT models is to generate more coherent and logically flowing events, rather than achieving precise state change modeling. These states act as contextual reminders within our environmental simulations, aiding the model in crafting new events. The core purpose of these states is to introduce variability in the events generated by the model, enabling it to anticipate "what might happen next" or consider "what unusual events could occur."
>
> One of the main challenges in our approach is not the model's inability to accurately simulate an environment, but rather its tendency to create overly perfect or neutral scenarios. The real world is in chaos. Users often perform seemingly random actions, such as clicking aimlessly, moving the mouse erratically, or hesitating over words they can't spell, leading to a plethora of confusing actions. Current models struggle to replicate this unpredictability.
>
> By incorporating detailed states and aligning generated events with real-world occurrences, we've found that our simulation environment can introduce a degree of uncertainty and reality. For instance:
>
> ```text
> {"time": "10-18 09:17:40", "content": "User inputs 'ctrl_l' with mouse interactions, clicks in an application."}
> {"time": "09-17 15:46:53", "content": "The user double-checks the warranty information for the shortlisted devices before making a purchase."}
> {"time": "10-05 14:26:56", "content": "The user uses 'ctrl+c' to copy data about material sustainability from EcoChic\u2019s website for documentation."}
> ```
>
> The final performance gain also demonstrates the effectiveness of our method.
>
> Last But Not Related To The Question: from our experience of trying to obtain events that simulate the real world, we assume that the final false alarm could be further reduced by incorporating more real-world chaos events into our training data. We also believe how to obtain random and uncertainty from models is a critical question to study in the future.

---

> ### Author Response · Authors · 2024-11-22
> **More tested baselines and broader applicability concerns; More examples in method part.**
>
> # W4. Concern about the tested baselines.
> To clarify, all test results shown in our paper are obtained with a chain-of-thought (CoT) method, as prompts in the appendix show.
> We here also present the results without CoT as a more naive baseline. This data is obtained with the following prompt:
>
> ```text
> <Role> You are a helpful assistant that provides proactive suggestions to the user. </Role>
>
> <Task> Understand what the user is doing and anticipate their needs based on events. Only propose assistance when you fully understand the user's actions. Use available operations to ensure the task is feasible. Execute the task if the user accepts your proposal. </Task>
>
> <Format> Respond in the following JSON format:
> {
>     "Proactive Task": "Describe your proposed task, or set to `null` if no assistance is needed.",
>     "Response": "Inform the user about your assistance if proposing a task."
> }
> </Format>
>
> <Rules>
> - Ensure the proposed task is relevant to the events. Focus on the user's current needs and predict helpful tasks.
> - Consider the timing of events.
> - Only offer proactive assistance when necessary.
> - Set `Proactive Task` to `null` if the user doesn't need help.
> </Rules>
> ```
>
> | Model        | Settings  | Recall  | Precision | Accuracy | False-Alarm | F1-Score |
> |--------------|-----------|---------|-----------|----------|-------------|----------|
> | GPT-4o-mini  | w/o CoT   | 100.00% | 18.83%    | 22.32%   | 81.17%      | 31.70%   |
> |              | w/  CoT   | 100.00% | 35.28%    | 36.12%   | 64.73%      | 52.15%   |
> | GPT-4o       | w/o CoT   | 88.06%  | 39.33%    | 57.51%   | 60.67%      | 54.38%   |
> |              | w/  CoT   | 98.11%  | 48.15%    | 49.78%   | 51.85%      | 64.60%   |
> | LLaMA-3.1-8B | w/o CoT   | 100.00% | 36.52%    | 37.34%   | 63.48%      | 53.50%   |
> |              | w/  CoT   | 98.86%  | 38.16%    | 39.06%   | 61.84%      | 55.06%   |
> |              | finetuned | 99.06%  | 49.76%    | 52.86%   | 50.24%      | 66.25%   |
>
> We can read from the result that after removing CoT, all models suffer from a performance drop.
> Besides, a comparison showing performance without any proactive agents is quite strange, as the main focus of this paper is to evaluate the proactiveness of the model.
>
>
> # W5. Concern about the broader applicability of the proactive agent approach.
>
> The proactive capability of agents is crucial as it allows systems to anticipate user needs and take preemptive actions, thereby enhancing efficiency and overall user experience. This capability is especially important in dynamic environments where timely and context-aware responses are essential.
>
> In our current study, we focus on general computer usage scenarios such as coding and writing. Our data generation methods can be easily adapted to broader scenarios by providing example events. However, due to the complexity of building monitoring software ( where we develop bundle of tools like VSCode add-ons, browser extensions, general user input monitoring, system monitoring, interactive panels) and the challenge of collecting and processing large volumes of data, we chose to focus on the three most common settings. Future work should leverage visual inputs to reduce the burden and extend to broader contexts.
> Here we would like to give more scenarios that require the proactive capability of agents:
>
> 1. Phone Assistance: By monitoring user actions and notifications, proactive agents can assist users in various ways, such as recommending searches for certain products based on family discussions or automatically scheduling calendar events by analyzing email content.
> 2. Healthcare Decision Support: In healthcare settings, the proactive agent can be employed to analyze patient medical records, symptoms, treatment history, and physiological data. It can predict and generate proactive actions such as recommending medical consultations, adjusting treatment plans, or managing hospital resources (e.g., bed adjustments) to enhance patient care.
> 3. Educational Tutoring Systems: Within educational environments, the proactive agent can record student interactions with learning materials, quiz responses, and progress over time. It can predict areas where students may need additional support and generate tailored learning resources or suggest targeted interventions to improve educational outcomes.
>
> We believe that the proactive agent has a broad range of application areas, making it a valuable tool for improving interactions and outcomes across numerous fields.
>
> # Q1. The suggestion of add more examples in the method part.
>
> We appreciate the suggestion to include more examples in the method section. To enhance clarity and understanding, we've added more examples in the method part and appendix.
>
> Thank you again for your dedication and effort :).

---

> ### Author Response · Authors · 2024-11-27
> **Request for Further Discussion Before Rebuttal Deadline**
>
> We have explained your concerns regarding the training process, dataset annotation, quality control, baseline, and potential applications. To improve clarity, we have also updated our paper. As the rebuttal deadline approaches, we welcome any additional feedback, suggestions, or concerns and are happy to engage in further discussion to clarify any remaining issues.

---

> ### Author Response · Authors · 2024-11-28
> **Follow up on Rebuttal Discussion and Further Feedback**
>
> We have made every effort to address your questions and concerns. As the rebuttal deadline approaches, we are eager to engage in a more in-depth discussion with you. We welcome any additional suggestions or unresolved issues you may have. Please feel free to reach out, and we will respond as quickly as possible.

---

> ### Author Response · Authors · 2024-11-29
> **Kindly Request for You Feedback Before Rebuttal Deadline**
>
> We have made every effort to clarify our points during the rebuttal period. However, we have not heard back from you since then. We kindly ask if there are any remaining issues or concerns that remain unresolved.
>
> As the deadline approaches, we would greatly appreciate your feedback or responses. Any additional concerns, suggestions, or questions are more than welcome, and we are eager to engage in further discussions to address them.
>
> Thank you very much for your time and thoughtful consideration.

---

> ### Author Response · Authors · 2024-12-02
> **Follow-Up on Rebuttal Feedback**
>
> We want to express our sincere gratitude for the time and effort you have dedicated to reviewing our paper. Your insights have been incredibly valuable to us.
>
> As the rebuttal deadline approaches, we wanted to kindly follow up regarding any remaining points or concerns you might have. We are eager to engage in further discussion and address any issues to improve our work.
>
> If you have any additional thoughts or suggestions, please feel free to share them with us. We are committed to providing prompt responses and appreciate your guidance in this process.
>
> Thank you once again for your consideration and support.

---

### Author Response · Authors · 2024-11-25
**Invitation for Further Discussion**

Thank you for your valuable feedback on our submission. We have carefully addressed each of your comments and believe further discussion could enhance our work. We invite you to review our responses and share any additional thoughts. Your input is invaluable in refining our research.

We look forward to your feedback and a constructive dialogue.

Thank you for your time and consideration.

---

### Author Response · Authors · 2024-12-04
**Brief Summary to Rebuttal**

Our work proposes a novel data-driven formalization for developing a proactive agent that anticipates user needs and takes the initiative by suggesting tasks or providing information without explicit requests.

Our contribution is as follows:
1. We designed an automatic data generation pipeline, which simulates user activities and responses to tasks predicted by proactive agents across various scenarios.
2. We proposed the benchmark ProactiveBench, which is used to evaluate and enhance the agent's proactive behavior. The dataset contains a total of 6790 events in the training set and 233 events in the test set.
3. We trained a reward model that achieved up to 91.80% consistency with human judgments in terms of F1-score to automatically evaluate the proactiveness of LLMs.

Final Score:
- Reviewer 9sAE: 6
- Reviewer ZTiB: 7
- Reviewer fw7i: 5
- Reviewer tUPL: 5

Reviewers Summary:
- Reviewer 9sAE, ZTiB, fw7i found the concept of **proactiveness** to be novel, interesting, important, and promising.
- Reviewer 9sAE, tUPL, fw7i recognized our work as **a valuable contribution** to the field, highlighting its potential to **benefit future research**.
- Reviewer tUPL, ZTiB regarded our framework as robust and logically sound, particularly emphasizing the **user agent design**, which effectively simulates real users to assess agent behavior and refine task propositions.

However, some reviewers raised concerns or sought clarification on specific aspects of our work:
- **Robustness of the benchmark**: Reviewer 9sAE, fw7i expressed concerns about the robustness of our benchmark due to its generation by GPT-4o. We argue that the reviewer fw7i might have some **misunderstanding** in our benchmark.  We clarified that our test data is a **replay of real-world events**, thoroughly explained the components of our dataset, and conducted additional experiments without Chain of Thought (CoT) to further validate the strength and robustness of our benchmark. Notably, GPT-4o improved its F1-score from **54.38% (w/o CoT)** to **64.60% (w/ CoT)**. Similarly, the fine-tuned model achieved the best overall performance, with an F1-score of **66.25%**.
- **Data annotation process**: Reviewer 9sAE, tUPL, ZTiB were interested in the details of the data annotation process. To address this, we added further explanations of human annotation procedures in our paper and provided additional clarifications in the rebuttal.
- **Model performance**: Reviewer tUPL, ZTiB sought more insights into the performance of different models. In response, we presented detailed case studies and highlighted **additional failure modes** to provide a comprehensive understanding of model performance. We also conducted new experiments for further analysis. Multi-rank predictions consistently outperform single-rank predictions. Specifically, GPT-4o improved its F1-score from **64.60% (pred@1)** to **77.72% (pred@3)**. Using a reward model will make the model propose less unnecessary predictions. E.g. GPT-4o improved its False-Alarm from **51.85% (pred@1)** to **44.74% (w/ RM)**. Combining both strategies will boost the model's performance. LLaMA-3.1-8B improved its F1-score from **44.74%(pred@1)** to **74.76%(pred@3 w/ RM)**.

---

### Meta-Review · Area_Chair_U51Y · 2024-12-20

**Metareview:**

(a) Summary: This paper proposes a novel data-driven formalization for developing a proactive agent that anticipates user needs and takes the initiative by suggesting tasks or providing information without explicit requests.
(b) Strengths: The paper is generally well-written and easy to follow. The proposed approach seems interesting and novel. The experimental results seem to support the authors' claims.
(c) Weaknesses: The reviewers pointed out a few concerns and issues. Some technical details need more explanations.
(d) This paper is a borderline case. Some of the reviewers replied to the authors' rebuttal and appreciated their refinement and clarifications. Generally speaking, I think that the authors addressed the majority of the reviewers' concerns and comments, although some of the reviewers did not reply during the rebuttal phase.

**Additional Comments On Reviewer Discussion:**

Some of the reviewers replied to the authors' rebuttal and appreciated their clarifications and refinement of the paper.

---

### Decision · Program_Chairs · 2025-01-22

Accept (Poster)